# Contrast Everything: A Hierarchical Contrastive Framework for Medical Time-Series

**Yihe Wang** *
University of North Carolina - Charlotte
ywang145@uncc.edu

**Yu Han**\*
University of Chinese Academy of Sciences
hanyu21@mails.ucas.ac.cn

**Haishuai Wang**
Zhejiang University
haishuai.wang@zju.edu.cn

**Xiang Zhang**
University of North Carolina - Charlotte
xiang.zhang@uncc.edu

## Abstract

Contrastive representation learning is crucial in medical time series analysis as it alleviates dependency on labor-intensive, domain-specific, and scarce expert annotations. However, existing contrastive learning methods primarily focus on one single data level, which fails to fully exploit the intricate nature of medical time series. To address this issue, we present COMET, an innovative hierarchical framework that leverages data consistencies at all inherent levels in medical time series. Our meticulously designed model systematically captures data consistency from four potential levels: observation, sample, trial, and patient levels. By developing contrastive loss at multiple levels, we can learn effective representations that preserve comprehensive data consistency, maximizing information utilization in a self-supervised manner. We conduct experiments in the challenging patient-independent setting. We compare COMET against six baselines using three diverse datasets, which include ECG signals for myocardial infarction and EEG signals for Alzheimer's and Parkinson's diseases. The results demonstrate that COMET consistently outperforms all baselines, particularly in setup with 10% and 1% labeled data fractions across all datasets. These results underscore the significant impact of our framework in advancing contrastive representation learning techniques for medical time series. The source code is available at https://github.com/DL4mHealth/COMET.

## 1 Introduction

Time series data is crucial in various real-world applications, ranging from finance [1, 2, 3], engineering [4, 5], to healthcare [6, 7]. Unlike domains such as computer vision [8, 9] and natural language processing [10, 11], where human recognizable features exist, time series data often lacks readily discernible patterns, making data labeling challenging. Consequently, the scarcity of labeled data poses a significant hurdle in effectively utilizing time series for analysis and classification tasks.

To address the paucity of labeled data in time series analysis, self-supervised contrastive learning has emerged as a promising approach. For example, TimeCLR [12] proposes a DTW data augmentation for time series data; TS2vec [13] designs a cropping and masking mechanism to form positive pairs; ExpCLR [14] introduces a novel loss function to utilize continuous expert features. By leveraging the inherent consistency within unlabeled data, contrastive learning algorithms enable the extraction of effective representations without relying on explicit labels. This paradigm shift opens up possibilities for overcoming the data scarcity issue and enhancing the capabilities of time series analysis.

---

*These authors contributed equally to this work.

37th Conference on Neural Information Processing Systems (NeurIPS 2023).

Despite recent advancements in contrastive learning methods for time series, existing approaches fail to exploit the full potential of medical time series data, such as electroencephalogram (EEG) signals. Unlike conventional time series data, medical time series often exhibit more data levels (Figure 1), including patient, trial, sample, and observation levels. Current contrastive learning techniques exclusively employ subsets of these levels (as illustrated in Table 1). Additionally, many of these methods are tailored to specific data types, which restricts their capacity to capture the rich complexity of medical time series. For example, CLOCS [15] presents a contrastive learning method for ECG using sample and patient levels. Mixing-up [16] captures sample-level consistency through a mixing data augmentation scheme. TNC [17] exploits trial-level consistency by contrasting neighbor samples in the same trial as positive pairs. Neither of them leverages all the levels exhibited in the medical time series.

After reviewing existing contrastive learning methods within the time series domain, we consistently posed a pivotal question to ourselves: Can we design a straightforward yet appliable contrastive learning framework that can be adapted to all forms of medical time series data, akin to the classical model SimCLR [18] in the domain of contrastive learning? Our objective is to craft an innovative framework that utilizes all information within medical time series in the context of self-supervised contrastive learning. It enables us to harness patient and trial information to learn consistency across instances while leveraging the sample and observation levels' information to facilitate conventional instance discrimination.

In this paper, we propose a hierarchical framework, COMET, that systematically leverages all four levels of medical time series, namely patient, trial, sample, and observation, to reduce the reliance on labeled data. By incorporating self-supervised contrastive learning, our method aims to bridge the gap between the limited availability of labeled data and the need for robust and generalizable models in medical time series analysis. We conduct extensive experiments with six baselines on three diverse datasets in a challenging patient-independent setting. COMET outperforms SOTAs by 14% and 13% F1 score with la-

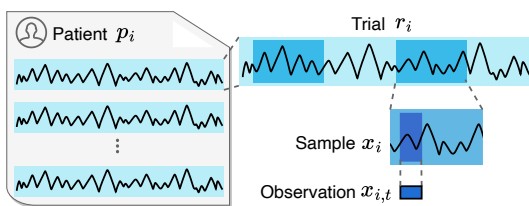

**Figure 1: Structure of medical time series.** Medical time series commonly have four levels (coarse to fine): patient, trial, sample, and observation. An observation is a single value in univariate time series and a vector in multivariate time series.

bel fractions of 10% and 1%, respectively, on EEG-based Alzheimer's detection. Further, COMET outperforms SOTAs by 0.17% and 2.66% F1 score with label fractions of 10% and 1%, respectively, on detecting Myocardial infarction with ECG. Finally, COMET outperforms SOTAs by 2% and 8% F1 score with label fractions of 10% and 1%, respectively, in the EEG-based diagnosis of Parkinson's disease. The results of downstream tasks demonstrate the effectiveness and stability of our method.

## 2 Related Work

**Medical time series.** Medical time series [19, 20, 21, 22] is a distinct type of time series data used for healthcare (*e.g.*, disease diagnosis, monitoring, and rehabilitation). It can be collected in a low-cost, non-invasive manner [23] or a high-cost, invasive manner [24].

Unlike general time series, which typically consist of sample and observation levels, medical time series introduces two additional data levels: patient and trial. These extra levels of data information in medical time series enable the development of specialized methods tailored to address the unique characteristics and requirements of medical time series analysis. Various types of medical time series, including EEG [28, 29, 30], ECG [31, 32], EMG [33], and EOG [34], offer valuable insights into specific medical conditions and play a crucial role in advancing healthcare practices.

**Table 1: Existing methods only utilize partial levels.**

| Models | Patient | Trial | Sample | Observation |
|---|---|---|---|---|
| **SimCLR [25]** | | | ✓ | |
| **TF-C [26]** | | | ✓ | |
| **Mixing-up [16]** | | | ✓ | |
| **TNC [17]** | | ✓ | | |
| **TS2vec [13]** | | | ✓ | ✓ |
| **TS-TCC [27]** | | | ✓ | ✓ |
| **CLOCS [15]** | ✓ | | ✓ | |
| **COMET(Ours)** | ✓ | ✓ | ✓ | ✓ |

**Contrastive learning for time series.** Contrastive learning has demonstrated its ability to learn effective representations in various domains, including image processing [35, 18, 36], graph analysis [37, 38, 37], and time series [39]. The key idea behind contrastive representation learning is to mine data consistency by bringing similar data closer together and pushing dissimilar data further apart.

Many existing works on contrastive learning focus on general time series, while some are designed specifically for medical data [40]. One classic framework, SimCLR, transforms a single sample into two augmented views and performs contrastive learning [25]. Other settings, such as using sub-series and overlapping-series, leverage sample-level consistency [41]. TF-C [26] contrasts representations learned from the time and frequency domains to exploit sample-level consistency. Mixing-up [16] learn sample-level consistency by utilizing a mixing component as a data augmentation scheme. TS-TCC and TS2vec [27, 13] apply data augmentation at the sample-level and perform contrastive learning at the observation-level. TNC [17] learns trial-level consistency by contrasting neighboring and non-neighboring samples. NCL [42] can also be used to learn trial-level consistency if we define samples from a trial as a neighborhood. CLOCS [15] learns patient-level consistency in cardiac signal features by contrasting different leads over time.

Certain prior methods have implicitly utilized hierarchical structure [13, 15]. However, as shown in Table 1, none of these methods leverage all the levels present in the medical time series, potentially resulting in the loss of useful information during training. We explicitly present a hierarchical framework in the context of contrastive learning, which can be applied across diverse types of medical time series data. In our paper, we aim to leverage data consistency at all levels in medical time series. Our work plays a role in summarizing, inspiring, and guiding future works in self-supervised contrastive learning on medical time series.

## 3 Preliminaries and Problem Formulation

### 3.1 Medical Time Series

In this section, we clarify the key conceptions of **observation** (or measurement), **sample** (or segment), **trial** (or recording), and **patient** (or subject) in the context of medical time series (Figure 1). For better understanding, we illustrate the concepts with an example of Electroencephalography (EEG) signals for Alzheimer's Disease diagnosis (details in Appendix A).

**Definition 1: Observation.** *An observation $\boldsymbol{x}_{i,t} \in \mathbb{R}^F$ in medical time series data represents a single data point or a vector captured at a specific timestamp $t$.* Here we use $i$ to denote the sample index (see **Definition 2**) and $t$ to denote the timestamp. It may record physiological status, laboratory test results, vital signs, or other measurable health indicators. The observation is a single real value for univariate time series while vector for multivariate time series. The $F$ is the feature dimension if it is a multivariate time series.

**Definition 2: Sample.** *A sample $\boldsymbol{x}_i = \{\boldsymbol{x}_{i,t} | t = 1, \cdots, T\}$ is a sequence of consecutive observations*, typically measured at regular intervals over a specified period ($T$ timestamps). It can also be called a *segment* or *window*. Here we use $i$ to denote the sample index. In the medical time series, a sample might consist of a sequence of heart rate measurements or blood pressure readings.

**Definition 3: Trial.** *A trial $\boldsymbol{r}_i$ is a collection of consecutive samples.* It can also be called a *record*. Here we use $i$ to denote the trial ID. In medical time series, a trial is a continuous set of observations collected over a not-short period (*e.g.*, 30 minutes). Therefore, a trial is generally too long (*e.g.*, hundreds of thousands of observations) to feed into deep learning models for representation learning directly and is usually split into shorter subsequences (*i.e.*, samples/segments). To represent the aggregate of samples stemming from a particular trial $\boldsymbol{r}_i$ with trial ID $i$, we employ the notation $\mathcal{R}_i$.

**Definition 4: Patient.** *A patient $\boldsymbol{p}_i$ represents a collection of multiple trials stemming from a single patient.* It can also be called a *subject*. Here we use $i$ to denote the patient ID. It is important to note that trials for a given patient may exhibit variations due to differing data collection timeframes, sensor placements, patient conditions, and other contributing factors. As shown in **Definition 3**, a trial is typically divided into many samples for better representation learning. In practical scenarios, a patient, which constitutes a cluster of trials, is also divided into samples that may share identical or distinct trial IDs but maintain the same patient ID. To represent the aggregate of samples stemming from a particular patient $\boldsymbol{p}_i$ with the corresponding patient ID $i$, we employ the notation $\mathcal{P}_i$.

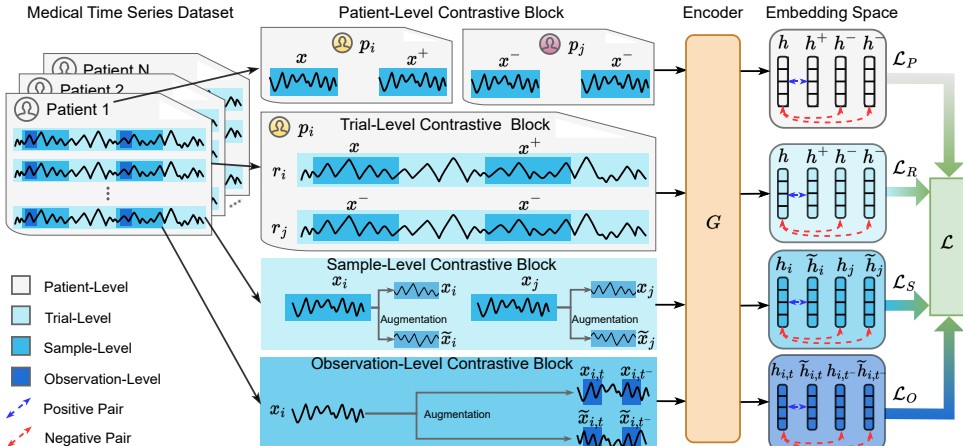

**Figure 2: Overview of COMET approach.** Our COMET model consists of four contrastive blocks, each illustrating the formulation of positive pairs and negative pairs at different data levels. In the observation-level contrastive, an observation $\boldsymbol{x}_{i,t}$ and its augmented view $\widetilde{\boldsymbol{x}}_{i,t}$ serve as a positive pair. Similarly, in the sample-level contrastive, a sample $\boldsymbol{x}_i$ and its augmented view $\widetilde{\boldsymbol{x}}_i$ form a positive pair. Moving to the trial-level contrastive, two samples $\boldsymbol{x}$ and $\boldsymbol{x}^+$ from the same trial $\boldsymbol{r}_i$ are considered to be a positive pair. The patient-level contrastive follows a similar pattern, where two samples $\boldsymbol{x}$ and $\boldsymbol{x}^+$ from the same patient $\boldsymbol{p}_i$ are regarded as a positive pair. Positive and corresponding negative pairs will be utilized to build contrastive loss in embedding space after being processed by encoder $G$.

In this work, we propose a novel hierarchical contrastive framework to learn representative and generalizable embeddings by comprehensively exploring instance discrimination at observation and sample levels and harnessing consistency across instances at trial and patient levels. Although we elaborate the proposed COMET in the context of medical time series, we note our model can possibly be extended to other time series beyond healthcare as long as extra information is available. For example, a climate dataset contains multiple meteorological satellites, each satellite contains multiple measuring units, and each unit contains multiple sensors, and every sensor can measure a specific observation at a certain timestamp. The key is to utilize all available information, excluding label data, for contrastive pre-training, such as patient ID. To adapt our approach to other domains, researchers must consider a crucial question: Does the dataset have additional information beyond sample labels? If affirmative, can this information be harnessed for contrastive learning? The example of satellite sensor application underscores the potential existence of supplementary information even in non-medical domains.

## 3.2 Problem Formulation

**Problem (Self-Supervised Representation Learning For Medical Time Series).** *Let an unlabeled dataset $\mathcal{D}$ consist of a set of patients, where each patient $\boldsymbol{p}_i$ has multiple trials, each trial $\boldsymbol{r}_i$ can be segmented into many samples, and each sample $\boldsymbol{x}_i$ comprises a series of observations. We aim to pre-train an encoder $G$ that exploits data consistency at all available levels in a self-supervised contrastive manner. For a given time series sample $\boldsymbol{x}_i \in \mathbb{R}^{T \times F}$ with $T$ timestamps and $F$ feature dimensions, the encoder $G$ learns a sample-level representation $\boldsymbol{h}_i \in \mathbb{R}^{T \times K}$, where $\boldsymbol{h}_{i,t} \in \mathbb{R}^K$ is the observation-level representation at timestamp $t$ with $K$ dimensions.*

By exploiting hierarchical consistency at multiple data levels, we aim to learn a representation $\boldsymbol{h}_i$ that is both representative (yielding good performance in downstream tasks) and generalizable (maintaining stability across different patients). Depending on the fine-tuning settings [18], a specific fraction of labels $y_i$ corresponding to samples $\boldsymbol{x}_i$ are necessary.

## 4 Method

In this section, we first present our assumption of data consistency behind designing a hierarchical contrastive framework. Then, we describe the architecture of the proposed model COMET (Figure 2).

## 4.1 Hierarchical Data Consistency

Capturing data consistency is crucial in the development of a contrastive learning framework [13]. Data consistency refers to the shared commonalities preserved within the data, which provide a supervisory signal to guide model optimization. Contrastive learning captures data consistency by contrasting positive and negative data pairs, where positive pairs share commonalities and negative pairs do not. We propose consistency across four data levels: observation, sample, trial, and patient, from fine-grained to coarse-grained in the medical time series. Although we present four levels here, our model can easily be adapted to accommodate specific datasets by adding or removing data levels.

**Observation-level data consistency.** We assume a slightly augmented observation (*e.g.*, channel masked) will carry similar information as the original observation [13]. We use $x_{i,t}$ as the anchor observation at timestamp $t$, and $x_{i,t^-}$ as the observation at another timestamp $t^-$ in the sample $x_i$. We consider the anchor observation $x_{i,t}$ and an augmented observation $\widetilde{x}_{i,t}$ as positive pairs $(x_{i,t}, \widetilde{x}_{i,t})$ (with closer embeddings). Conversely, we consider the original observation $x_{i,t}$ and the observations $\widetilde{x}_{i,t^-}$ and $x_{i,t^-}$ at another timestamp $t^-$ as negative pairs $(x_{i,t}, \widetilde{x}_{i,t^-}), (x_{i,t}, x_{i,t^-})$, with distant embeddings.

**Sample-level data consistency.** The sample-level consistency is based on our assumption that a slightly perturbed sample (e.g., temporally masked) should carry similar information as the original sample [18, 16, 26]. We consider the anchor sample $x_i$ and its augmented view $\widetilde{x}_i$ as positive pair $(x_i, \widetilde{x}_i)$. We regard the anchor sample $x_i$ and a different sample $x_j$ and its augmented view $\widetilde{x}_j$ as negative pairs: $(x_i, \widetilde{x}_j)$ and $(x_i, x_j)$.

**Trial-level data consistency.** We assume that samples sliced from the same trial should carry similar information compared to those obtained from different trials. For simplicity, we use $x$ to denote the anchor sample and $x^+$ to denote a sample from the same trial $r_i$ as the anchor sample, while $x^-$ to denote a sample from another trial $r_j$. In other words, we have $\{x, x^+\} \in \mathcal{R}_i$ and $x^- \in \mathcal{R}_j$. We treat sample $x$ and the sample $x^+$ from the same trial as positive pair $(x, x^+)$. We regard sample $x$ and the sample $x^-$ from different trials as negative pair $(x, x^-)$.

**Patient-level data consistency.** We assume samples originating from the same patient are likely to contain similar information when compared to those from different patients [15]. Here, we use $x$ to denote the anchor sample and $x^+$ to denote a sample from the same patient $p_i$, while $x^-$ from another patient $p_j$. In other words, there are $\{x, x^+\} \in \mathcal{P}_i$ and $x^- \in \mathcal{P}_j$. We have positive pair $(x, x^+)$ including samples from the same patient and negative pair $(x, x^-)$ that from different patients.

**Disease-level data consistency.** For completeness, we introduce disease-level data consistency, which suggests that samples associated with the same type of disease should exhibit shared patterns, even when collected from different patients in different ways. However, capturing disease-level consistency requires ground truth labels, which are not available in a self-supervised approach. As a result, we do NOT employ disease-level consistency in this paper. Nevertheless, it can be adapted for semi-supervised or supervised contrastive learning and may prove beneficial in learning domain-adaptable representations for certain diseases across patients and even datasets.

A common principle underlying all definitions is that *the X-level data consistency refers to the positive pair belonging to the same X, where X could be observation, sample, trial, patient, or disease.* We assume that *each patient is associated with only one label*, such as suffering from a specific disease, which implies that all samples from the same patient essentially originate from the same distribution. However, in cases where data from a patient is derived from multiple distributions (e.g., a patient could perform various daily activities; associated with multiple labels), the assumptions of trial-level and patient-level consistency are not satisfied. Therefore, the user can switch on the observation-level and sample-level consistency.

Building upon the concepts of data consistency, we introduce four contrastive blocks corresponding to the four data levels. Our model is highly flexible, allowing users to enable or disable any of the blocks based on the requirements of a specific task or dataset.

## 4.2 Observation-Level Contrastive Block

For a given time series sample $x_i$, we apply data augmentation (such as masking) to generate an augmented sample $\widetilde{x}_i$ [25, 43]. We input the original sample $x_i$ and its augmented view $\widetilde{x}_i$ into

contrastive encoder $G$ to obtain their respective representations $\boldsymbol{h}_i = G(\boldsymbol{x}_i)$ and $\widetilde{\boldsymbol{h}}_i = G(\widetilde{\boldsymbol{x}}_i)$. It is important to note that we apply data augmentation to the samples, which indirectly extends to augmenting the observations, simplifying the encoding process. To capture observation-level consistency, we assume that, after being processed by encoder $G$, the representation of observation $\boldsymbol{x}_{i,t}$ is close to the representation of the augmented observation $\widetilde{\boldsymbol{x}}_{i,t}$. In contrast, it should be distant from the representations of observations $\boldsymbol{x}_{i,t^-}$ and $\widetilde{\boldsymbol{x}}_{i,t^-}$ originating from any other timestamp $t^-$. Specifically, our positive pair is $(\boldsymbol{x}_{i,t}, \widetilde{\boldsymbol{x}}_{i,t})$ and negative pairs are $(\boldsymbol{x}_{i,t}, \boldsymbol{x}_{i,t^-})$ and $(\boldsymbol{x}_{i,t}, \widetilde{\boldsymbol{x}}_{i,t^-})$.

**Observation-level contrastive loss.** The observation-level contrastive loss $\mathcal{L}_O$ [13] for the input sample $\boldsymbol{x}_i$ is defined as:

$$\mathcal{L}_O = \mathbb{E}_{\boldsymbol{x}_i \in \mathcal{D}} \left[ \mathbb{E}_{t \in \mathcal{T}} \left[ -\log \frac{\exp(\boldsymbol{h}_{i,t} \cdot \widetilde{\boldsymbol{h}}_{i,t})}{\sum_{t^- \in \mathcal{T}} (\exp(\boldsymbol{h}_{i,t} \cdot \widetilde{\boldsymbol{h}}_{i,t^-}) + \mathbb{1}_{[t \neq t^-]} \exp(\boldsymbol{h}_{i,t} \cdot \boldsymbol{h}_{i,t^-}))} \right] \right] \quad (1)$$

where $\mathcal{T} = \{1, \cdots, T\}$ is the set of all timestamps in sample $\boldsymbol{x}_i$ and $\cdot$ denotes dot product. The $\mathbb{1}_{[t \neq t^-]}$ is an indicator function that equals to $0$ when $t = t^-$ and $1$ otherwise.

### 4.3 Sample-Level Contrastive Block

For an input time series sample $\boldsymbol{x}_i$ and its augmented view $\widetilde{\boldsymbol{x}}_i$, we calculate their representations through $\boldsymbol{h}_i = G(\boldsymbol{x}_i)$ and $\widetilde{\boldsymbol{h}}_i = G(\widetilde{\boldsymbol{x}}_i)$. The augmentation applied here could be the same as or different from the augmentation used in Section 4.2. We assume that after passing through the encoder $G$, the representation of the sample $\boldsymbol{x}_i$ is close to the representation of its augmented view $\widetilde{\boldsymbol{x}}_i$, while far away from the representations of any other samples $\boldsymbol{x}_j$ and $\widetilde{\boldsymbol{x}}_j$. In specific, our positive pair is $(\boldsymbol{x}_i, \widetilde{\boldsymbol{x}}_i)$, and negative pairs are $(\boldsymbol{x}_i, \widetilde{\boldsymbol{x}}_j)$ and $(\boldsymbol{x}_i, \boldsymbol{x}_j)$.

**Sample-level contrastive loss.** The sample-level contrastive loss $\mathcal{L}_S$ [18, 43] for the input sample $\boldsymbol{x}_i$ is defined as:

$$\mathcal{L}_S = \mathbb{E}_{\boldsymbol{x}_i \in \mathcal{D}} \left[ -\log \frac{\exp(\boldsymbol{h}_i \cdot \widetilde{\boldsymbol{h}}_i)}{\sum_{j=1}^{|\mathcal{D}|} (\exp(\boldsymbol{h}_i \cdot \widetilde{\boldsymbol{h}}_j) + \mathbb{1}_{[i \neq j]} \exp(\boldsymbol{h}_i \cdot \boldsymbol{h}_j))} \right] \quad (2)$$

where $|\mathcal{D}|$ represents the total number of samples in the dataset $\mathcal{D}$ and $\cdot$ denotes dot product. The $\mathbb{1}_{[i \neq j]}$ is an indicator function that equals $0$ when $i = j$ and $1$ otherwise.

### 4.4 Trial-Level Contrastive Block

For an input sample $\boldsymbol{x} \in \mathcal{R}_i$, where $\mathcal{R}_i$ is a collection of all samples segmented from trial $\boldsymbol{r}_i$, we feed it into the contrastive encoder $G$ to generate a sample-level representation $\boldsymbol{h} = G(\boldsymbol{x})$. To seize trial-level data consistency, we assume that the representation of the anchor sample $\boldsymbol{x} \in \mathcal{R}_i$ is close to the representation of sample $\boldsymbol{x}^+$ that also come from the trial $\boldsymbol{r}_i$. In contrast, the representation of the anchor sample $\boldsymbol{x}$ is far away from the representation of sample $\boldsymbol{x}^-$ that come from a different trial $\boldsymbol{r}_j$, where $\boldsymbol{x}^- \in \mathcal{R}_j$. In other words, we have positive pair $(\boldsymbol{x}, \boldsymbol{x}^+)$ and negative pair $(\boldsymbol{x}, \boldsymbol{x}^-)$.

**Trial-level contrastive loss.** The trial-level contrastive loss $\mathcal{L}_R$ [15, 18] for the input sample $\boldsymbol{x}$ is defined as:

$$\mathcal{L}_R = \mathbb{E}_{\boldsymbol{x} \in \mathcal{D}} \left[ \mathbb{E}_{\boldsymbol{x}^+ \in \mathcal{R}_i} \left[ -\log \frac{\exp(\text{sim}(\boldsymbol{h}, G(\boldsymbol{x}^+))/\tau)}{\sum_{j=1}^{J} \sum_{\boldsymbol{x}^- \in \mathcal{R}_j} (\exp(\text{sim}(\boldsymbol{h}, G(\boldsymbol{x}^-))/\tau))} \right] \right] \quad (3)$$

where $J$ is the total number of trials in the dataset $\mathcal{D}$. The $\text{sim}(\boldsymbol{u}, \boldsymbol{v}) = \boldsymbol{u}^T \boldsymbol{v} / \|\boldsymbol{u}\| \|\boldsymbol{v}\|$ denotes the cosine similarity, and $\tau$ is a temperature parameter to adjust the scale. The $G(\boldsymbol{x}^+)$ and $G(\boldsymbol{x}^-)$ are learned representations of samples $\boldsymbol{x}^+ \in \mathcal{R}_i$ and $\boldsymbol{x}^- \in \mathcal{R}_j$, respectively. To measure the trial-level loss for sample $\boldsymbol{x}$, we iterate all the $\boldsymbol{x}^+$ in $\mathcal{R}_i$, and averaging across $|\mathcal{R}_i| - 1$ positive pairs.

In this block, we do NOT learn a trial-level embedding representing the entire trial. Instead, we learn a representation for each sample within the trial while considering trial-level data consistency. Similarly, we follow this protocol for the patient-level contrastive block.

### 4.5 Patient-Level Contrastive Block

For an input sample $\boldsymbol{x} \in \mathcal{P}_i$, where $\mathcal{P}_i$ denotes all samples from patient $\boldsymbol{p}_i$, we feed it into the contrastive encoder $G$ to generate a sample-level representation $\boldsymbol{h} = G(\boldsymbol{x})$. Similar to the above

trial-level contrastive block, we have positive pair $(\boldsymbol{x}, \boldsymbol{x}^+)$ and negative pair $(\boldsymbol{x}, \boldsymbol{x}^-)$, in which $\boldsymbol{x}^+$ come from the same patient while $\boldsymbol{x}^-$ come from a different patient.

**Patient-level contrastive loss.** The patient-level contrastive loss $\mathcal{L}_\text{P}$ [15] for the input sample $\boldsymbol{x}$ is defined as:

$$\mathcal{L}_\text{P} = \mathbb{E}_{\boldsymbol{x} \in \mathcal{D}} \left[ \mathbb{E}_{\boldsymbol{x}^+ \in \mathcal{P}_i} \left[ -\log \frac{\exp(\text{sim}(\boldsymbol{h}, G(\boldsymbol{x}^+))/\tau)}{\sum_{j=1}^M \sum_{\boldsymbol{x}^- \in \mathcal{P}_j}(\exp(\text{sim}(\boldsymbol{h}, G(\boldsymbol{x}^-))/\tau))} \right] \right] \tag{4}$$

where $M$ is the total number of patients in the dataset $\mathcal{D}$. In this block, the $G(\boldsymbol{x}^+)$ and $G(\boldsymbol{x}^-)$ are learned representations of samples $\boldsymbol{x}^+ \in \mathcal{P}_i$ and $\boldsymbol{x}^- \in \mathcal{P}_j$, respectively.

### 4.6 Overall Loss Function

The overall loss function $\mathcal{L}$ consists of four loss terms The observation-level loss $\mathcal{L}_\text{O}$ and sample-level loss $\mathcal{L}_\text{S}$ encourage the encoder to learn robust representations that are invariant to perturbations. The trial-level loss $\mathcal{L}_\text{R}$ and patient-level loss $\mathcal{L}_\text{P}$ compel the encoder to learn cross-sample features within a trial or a patient. In summary, the overall loss function of the proposed COMET model is:

$$\mathcal{L} = \lambda_1 \mathcal{L}_\text{O} + \lambda_2 \mathcal{L}_\text{S} + \lambda_3 \mathcal{L}_\text{R} + \lambda_4 \mathcal{L}_\text{P} \tag{5}$$

where $\lambda_1, \lambda_2, \lambda_3, \lambda_4 \in [0, 1]$ are hyper-coefficients that control the relative importance and adjust the scales of each level's loss. Users can simply turn off specific data levels by setting $\lambda$ of those levels to 0. We set $\lambda_1 + \lambda_2 + \lambda_3 + \lambda_4 = 1$. We calculate the total loss by taking the expectation of $\mathcal{L}$ across all samples $\boldsymbol{x} \in \mathcal{D}$. In practice, the contrastive losses are calculated within a mini-batch.

## 5 Experiments

We compare the COMET model with six baselines on three datasets. Following the setup in SimCLR [18], we use unlabeled data to pre-train encoder $G$ and evaluate it in two downstream settings (Appendix E): partial fine-tuning (P-FT; *i.e.*, linear evaluation [18]) and full fine-tuning (F-FT). All datasets are split into training, validation, and test sets in **patient-independent** setting (Figure 3), which is very challenging due to patient variations (More explanations in Appendix F.1).

**Datasets.** (1) **AD** [44] has 23 patients, 663 trials, and 5967 multivariate EEG samples. There are 4329, 891, and 747 samples in training, validation, and test sets. The sampling rate (frequency) is 256Hz. Each sample is a one-second interval with 256 observations. A binary label based on whether the patient has Alzheimer's disease is assigned to each sample. (2) **PTB** [45] has 198 patients, 6237 trials, and 62370 multivariate ECG samples. There are 53950, 3400,

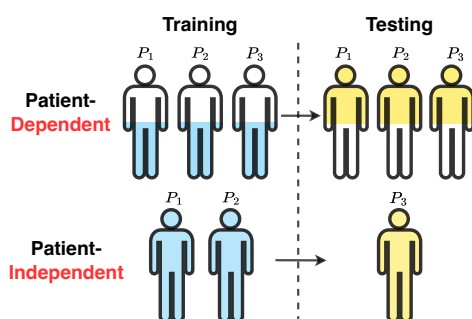

**Figure 3: Patient-dependent/independent Setting.** In the patient-dependent setting, samples from the same patient can appear in both the training and test sets. In contrast, in the patient-independent setting, samples from the same patient are exclusively included in the training or test set.

and 5020 samples in training, validation, and test sets. The sampling rate (frequency) is 250Hz. Each sample is a heartbeat with 300 observations. A binary label based on whether the patient has Myocardial infarction is assigned to each sample. (3) **TDBRAIN** [46] has 72 patients, 624 trials, and 11856 multivariate EEG samples. There are 8208, 1824, and 1824 samples in training, validation, and test sets. The sampling rate (frequency) is 256Hz. Each sample is a one-second interval with 256 observations. A binary label based on whether the patient has Parkinson's disease is assigned to each sample. See appendix D for more details about data statistics, train-test split, and data preprocessing.

**Baselines.** We compare with 6 state-of-the-art methods: TS2vec [13], Mixing-up [16], TS-TCC [27], SimCLR [43], CLOCS [15] and TF-C [26]. Since TF-C is designed for transfer learning, we implement its downstream tasks the same as ours for a fair comparison. The evaluation metrics are accuracy, precision (macro-averaged), recall(macro-averaged), F1 score(macro-averaged, AUROC(macro-averaged), and AUPRC(macro-averaged).

**Table 2: Partial fine-tuning results**. A logistic regression(LG) classifier $L$ is added on top of a frozen encoder $G$ on dataset AD. Only fine-tuning the classifier $L$.

| Models | Accuracy | Precision | Recall | F1 score | AUROC | AUPRC |
|--------|----------|-----------|--------|----------|-------|-------|
| TS2vec | $66.48_{\pm5.33}$ | $67.72_{\pm5.09}$ | $67.40_{\pm5.20}$ | $66.32_{\pm5.46}$ | $74.12_{\pm6.88}$ | $72.96_{\pm7.21}$ |
| TF-C | $77.03_{\pm2.80}$ | $75.79_{\pm5.07}$ | $64.27_{\pm5.03}$ | $64.85_{\pm5.56}$ | $80.71_{\pm4.03}$ | $79.27_{\pm4.15}$ |
| Mixing-up | $46.16_{\pm1.38}$ | $52.62_{\pm4.90}$ | $50.81_{\pm1.32}$ | $37.37_{\pm1.98}$ | $64.42_{\pm6.49}$ | $62.85_{\pm6.07}$ |
| TS-TCC | $59.71_{\pm8.63}$ | $61.66_{\pm8.63}$ | $60.33_{\pm8.26}$ | $58.66_{\pm8.39}$ | $67.53_{\pm10.04}$ | $68.33_{\pm9.37}$ |
| SimCLR | $57.16_{\pm2.05}$ | $56.67_{\pm3.91}$ | $53.57_{\pm2.21}$ | $49.11_{\pm4.26}$ | $56.67_{\pm3.91}$ | $52.10_{\pm1.41}$ |
| CLOCS | $66.99_{\pm2.84}$ | $67.17_{\pm2.96}$ | $67.33_{\pm2.99}$ | $66.91_{\pm2.88}$ | $73.71_{\pm3.62}$ | $72.58_{\pm3.54}$ |
| **COMET (Ours)** | $76.09_{\pm4.21}$ | $\mathbf{77.36_{\pm3.97}}$ | $\mathbf{74.68_{\pm4.62}}$ | $\mathbf{74.80_{\pm4.83}}$ | $\mathbf{81.30_{\pm4.97}}$ | $\mathbf{80.50_{\pm5.31}}$ |

**Implementation.** We use a ten residual blocks dilated CNN module [47] as backbones for encoders $G$. To preserve the magnitude feature of time series, which is often an important feature in time series, we utilize timestamp masking as proposed in [13] as an augmentation method. While many existing works apply data augmentation directly on raw time series, we use a fully connected projection layer to map the raw input into an embedding with a higher feature dimension. This strategy helps prevent situations where some parts of the raw data are zero, which could result in ineffective augmentation if timestamp masking were directly applied. Additionally, to ensure stability during training, we incorporate a hierarchical loss [13], which encapsulates the observation and sample-level losses. To guarantee there are samples with the same trial and patient IDs in a batch for trial and patient levels contrasting, we design two specific shuffle functions. See Appendix C for further details. For datasets whose trial level is absent or limited to one trial per patient, we provide a solution in Appendix F.2.

We conduct experiments with five seeds(41-45) based on the same data split to account for variance and report the mean and standard deviation. All experiments expect baseline SimCLR run on NVIDIA RTX 4090. Baseline SimCLR runs on Google Colab NVIDIA A100. See further implementation details in Appendix E.

## 5.1 Results on Partial Fine-tuning

**Setup.** Linear evaluation is a widely used P-FT setup to evaluate the quality of representation, where a linear classifier is trained on top of a frozen network for sample classification [18, 15, 48, 49]. This evaluation method allows for a quick assessment of representation quality. In linear evaluation, we add a logistic regression classifier $L$ on top of the pre-trained encoder $G$. The training process utilizes 100% labeled training data for sample classification. Notably, the parameters of the encoder $G$ are frozen during training, ensuring that only the classifier $L$ is fine-tuned. See further details in E.1.

**Results.** We report the experimental results of the P-FT setup on AD in Table 2. On average, our COMET model claims a large margin of more than 7% over all baselines on the F1 score on AD.

## 5.2 Results on Full Fine-tuning

**Setup.** In an F-FT setup, we add a two-layer fully connected network classifier $P$ to the pre-trained encoder $G$. The training process utilizes 100%, 10%, and 1% of the labeled training data for sample classification, respectively. Unlike the P-FT approach, both the encoder $G$ and classifier $P$ are trainable now, allowing for fine-tuning of the entire network structure. See further details in E.2.

**Results.** The results for F-FT on the three datasets are presented in Table 3 and Table 4. In general, COMET demonstrates success in 46 out of 54 tests conducted across the three datasets, considering six different evaluation metrics. With 100% labeled data, the F1 score of COMET outperforms the best baseline, TS2Vec, by 2% on the AD dataset and surpasses the best baseline, Mixing-up, by 4% on the TDBrain dataset. Furthermore, it achieves a result comparable to the best baseline, TF-C, on the PTB dataset.

Notably, the F1 score of COMET surpasses the best baseline, TF-C, by 14% and 13% with label fractions of 10% and 1%, respectively, on the AD dataset. Additionally, on the TDBrain dataset, the F1 score of COMET outperforms the best baseline, Mixing-up, by 2% and 8% with label fractions of 10% and 1%, respectively. Similarly, the F1 score of COMET outperforms the best baseline, CLOCS, by 0.17% and 2.66% with label fractions of 10% and 1%, respectively, on the PTB dataset. It is interesting to observe that the F1 score of COMET with label fractions of 10% and 1% outperforms a label fraction of 100% on the AD and PTB datasets. This suggests a potential overfitting of COMET to the training data. A similar phenomenon is observed with SimCLR, TF-C, CLOCS, and TS2vec, where a higher fraction of labeled data did not necessarily lead to improved performance.

Table 3: **Full fine-tuning results of EEG datasets**. A two-layer fully connected networks classifier $P$ is trained on top of pre-trained encoder $G$ on EEG datasets AD and TDBRAIN. Fine-tuning the whole network includes $G$ and $P$. We use 100%, 10%, and 1% of labeled training data in AD and TDBRAIN.

| Datasets | Fraction | Models | Accuracy | Precision | Recall | F1 score | AUROC | AUPRC |
|---|---|---|---|---|---|---|---|---|
| AD | 100% | TS2vec | $81.26_{\pm 2.08}$ | $81.21_{\pm 2.14}$ | $81.34_{\pm 2.04}$ | $81.12_{\pm 2.06}$ | $89.20_{\pm 1.76}$ | $88.94_{\pm 1.85}$ |
| | | TF-C | $75.31_{\pm 8.27}$ | $75.87_{\pm 8.73}$ | $74.83_{\pm 8.98}$ | $74.54_{\pm 8.85}$ | $79.45_{\pm 10.23}$ | $79.33_{\pm 10.57}$ |
| | | Mixing-up | $65.68_{\pm 7.89}$ | $72.61_{\pm 4.21}$ | $68.25_{\pm 6.97}$ | $63.98_{\pm 9.92}$ | $84.63_{\pm 5.04}$ | $83.46_{\pm 5.48}$ |
| | | TS-TCC | $73.55_{\pm 10.00}$ | $77.22_{\pm 6.13}$ | $73.83_{\pm 9.65}$ | $71.86_{\pm 11.59}$ | $86.17_{\pm 5.11}$ | $85.73_{\pm 5.11}$ |
| | | SimCLR | $54.77_{\pm 1.97}$ | $50.15_{\pm 7.02}$ | $50.58_{\pm 1.92}$ | $43.18_{\pm 4.27}$ | $50.15_{\pm 7.02}$ | $50.42_{\pm 1.06}$ |
| | | CLOCS | $78.37_{\pm 6.00}$ | $83.99_{\pm 2.11}$ | $76.14_{\pm 7.03}$ | $75.78_{\pm 7.93}$ | $91.17_{\pm 2.51}$ | $90.72_{\pm 3.05}$ |
| | | **COMET (Ours)** | $\mathbf{84.50_{\pm 4.46}}$ | $\mathbf{88.31_{\pm 2.42}}$ | $\mathbf{82.95_{\pm 5.39}}$ | $\mathbf{83.33_{\pm 5.15}}$ | $\mathbf{94.44_{\pm 2.37}}$ | $\mathbf{94.43_{\pm 2.48}}$ |
| | 10% | TS2vec | $73.28_{\pm 4.34}$ | $74.14_{\pm 4.33}$ | $73.52_{\pm 3.77}$ | $73.00_{\pm 4.18}$ | $81.66_{\pm 5.20}$ | $81.58_{\pm 5.11}$ |
| | | TF-C | $75.66_{\pm 11.21}$ | $75.48_{\pm 11.48}$ | $75.58_{\pm 11.59}$ | $75.38_{\pm 11.47}$ | $81.38_{\pm 14.19}$ | $81.56_{\pm 13.68}$ |
| | | Mixing-up | $59.38_{\pm 3.33}$ | $64.85_{\pm 4.38}$ | $61.94_{\pm 3.42}$ | $58.17_{\pm 3.41}$ | $75.02_{\pm 6.14}$ | $73.44_{\pm 5.82}$ |
| | | TS-TCC | $77.83_{\pm 6.90}$ | $79.73_{\pm 7.49}$ | $76.18_{\pm 7.21}$ | $76.43_{\pm 7.56}$ | $84.12_{\pm 7.32}$ | $84.12_{\pm 7.61}$ |
| | | SimCLR | $56.09_{\pm 2.25}$ | $53.81_{\pm 5.74}$ | $51.73_{\pm 2.59}$ | $44.10_{\pm 4.84}$ | $53.81_{\pm 5.74}$ | $51.08_{\pm 1.53}$ |
| | | CLOCS | $76.97_{\pm 3.01}$ | $81.70_{\pm 3.21}$ | $74.69_{\pm 3.26}$ | $74.75_{\pm 3.61}$ | $86.91_{\pm 3.61}$ | $86.70_{\pm 3.64}$ |
| | | **COMET (Ours)** | $\mathbf{91.43_{\pm 3.12}}$ | $\mathbf{92.52_{\pm 2.36}}$ | $\mathbf{90.71_{\pm 3.56}}$ | $\mathbf{91.14_{\pm 3.31}}$ | $\mathbf{96.44_{\pm 2.84}}$ | $\mathbf{96.48_{\pm 2.82}}$ |
| | 1% | TS2vec | $64.93_{\pm 3.53}$ | $65.28_{\pm 3.52}$ | $65.14_{\pm 3.59}$ | $64.64_{\pm 3.58}$ | $70.56_{\pm 5.38}$ | $68.97_{\pm 5.75}$ |
| | | TF-C | $75.66_{\pm 12.61}$ | $75.26_{\pm 13.91}$ | $74.77_{\pm 13.64}$ | $74.33_{\pm 14.45}$ | $78.96_{\pm 16.17}$ | $81.89_{\pm 12.50}$ |
| | | Mixing-up | $63.67_{\pm 1.20}$ | $65.02_{\pm 2.09}$ | $64.64_{\pm 1.63}$ | $63.55_{\pm 1.21}$ | $71.95_{\pm 3.39}$ | $70.15_{\pm 3.70}$ |
| | | TS-TCC | $53.04_{\pm 8.80}$ | $52.39_{\pm 13.73}$ | $52.00_{\pm 8.01}$ | $44.69_{\pm 12.24}$ | $48.89_{\pm 10.59}$ | $51.41_{\pm 7.81}$ |
| | | SimCLR | $55.42_{\pm 2.43}$ | $52.18_{\pm 5.55}$ | $51.37_{\pm 2.76}$ | $45.02_{\pm 4.79}$ | $52.18_{\pm 5.55}$ | $50.87_{\pm 1.45}$ |
| | | CLOCS | $64.50_{\pm 4.16}$ | $65.17_{\pm 4.72}$ | $63.73_{\pm 4.86}$ | $63.01_{\pm 5.11}$ | $69.16_{\pm 6.75}$ | $68.15_{\pm 7.20}$ |
| | | **COMET (Ours)** | $\mathbf{88.22_{\pm 3.36}}$ | $\mathbf{88.55_{\pm 2.73}}$ | $\mathbf{88.56_{\pm 3.14}}$ | $\mathbf{88.14_{\pm 3.37}}$ | $\mathbf{96.05_{\pm 1.36}}$ | $\mathbf{96.12_{\pm 1.31}}$ |
| TDBrain | 100% | TS2vec | $80.21_{\pm 1.69}$ | $81.38_{\pm 1.97}$ | $80.21_{\pm 1.69}$ | $80.07_{\pm 1.69}$ | $89.57_{\pm 2.31}$ | $89.60_{\pm 2.37}$ |
| | | TF-C | $66.62_{\pm 1.76}$ | $67.15_{\pm 1.64}$ | $66.62_{\pm 1.76}$ | $66.35_{\pm 1.91}$ | $65.43_{\pm 6.13}$ | $66.18_{\pm 4.90}$ |
| | | Mixing-up | $81.47_{\pm 1.07}$ | $82.11_{\pm 1.12}$ | $81.47_{\pm 1.07}$ | $81.38_{\pm 1.08}$ | $90.48_{\pm 0.89}$ | $90.51_{\pm 0.94}$ |
| | | TS-TCC | $77.42_{\pm 2.86}$ | $77.68_{\pm 2.93}$ | $77.42_{\pm 2.86}$ | $77.37_{\pm 2.86}$ | $87.37_{\pm 3.06}$ | $87.61_{\pm 3.22}$ |
| | | SimCLR | $58.43_{\pm 1.77}$ | $59.48_{\pm 1.95}$ | $58.43_{\pm 1.77}$ | $57.30_{\pm 2.07}$ | $59.48_{\pm 1.95}$ | $55.05_{\pm 1.18}$ |
| | | CLOCS | $78.16_{\pm 1.13}$ | $79.49_{\pm 1.61}$ | $78.16_{\pm 1.13}$ | $77.91_{\pm 1.12}$ | $88.49_{\pm 1.95}$ | $88.83_{\pm 1.94}$ |
| | | **COMET (Ours)** | $\mathbf{85.47_{\pm 1.16}}$ | $\mathbf{85.68_{\pm 1.20}}$ | $\mathbf{85.47_{\pm 1.16}}$ | $\mathbf{85.45_{\pm 1.16}}$ | $\mathbf{93.73_{\pm 1.02}}$ | $\mathbf{93.96_{\pm 0.99}}$ |
| | 10% | TS2vec | $72.39_{\pm 1.13}$ | $74.49_{\pm 1.73}$ | $72.39_{\pm 1.13}$ | $71.80_{\pm 1.05}$ | $80.71_{\pm 1.90}$ | $80.06_{\pm 1.87}$ |
| | | TF-C | $59.14_{\pm 3.04}$ | $59.34_{\pm 3.19}$ | $59.14_{\pm 3.04}$ | $58.98_{\pm 2.94}$ | $59.56_{\pm 4.10}$ | $59.65_{\pm 2.99}$ |
| | | Mixing-up | $77.50_{\pm 2.07}$ | $\mathbf{80.09_{\pm 1.92}}$ | $77.50_{\pm 2.07}$ | $76.99_{\pm 2.28}$ | $87.29_{\pm 1.34}$ | $87.13_{\pm 1.37}$ |
| | | TS-TCC | $71.23_{\pm 1.57}$ | $78.78_{\pm 0.66}$ | $71.23_{\pm 1.57}$ | $69.18_{\pm 2.25}$ | $80.56_{\pm 1.98}$ | $80.21_{\pm 2.21}$ |
| | | SimCLR | $59.79_{\pm 2.09}$ | $60.50_{\pm 1.90}$ | $59.79_{\pm 2.09}$ | $59.06_{\pm 2.82}$ | $60.50_{\pm 1.90}$ | $55.96_{\pm 1.41}$ |
| | | CLOCS | $75.04_{\pm 2.65}$ | $75.97_{\pm 2.86}$ | $75.04_{\pm 2.65}$ | $74.83_{\pm 2.66}$ | $84.25_{\pm 3.27}$ | $84.37_{\pm 3.52}$ |
| | | **COMET (Ours)** | $\mathbf{79.28_{\pm 4.64}}$ | $79.83_{\pm 4.83}$ | $\mathbf{79.28_{\pm 4.64}}$ | $\mathbf{79.19_{\pm 4.62}}$ | $\mathbf{88.39_{\pm 4.13}}$ | $\mathbf{88.38_{\pm 3.96}}$ |
| | 1% | TS2vec | $58.16_{\pm 2.30}$ | $58.47_{\pm 2.48}$ | $58.16_{\pm 2.30}$ | $57.83_{\pm 2.19}$ | $61.26_{\pm 2.51}$ | $60.95_{\pm 2.48}$ |
| | | TF-C | $53.16_{\pm 2.11}$ | $53.19_{\pm 2.10}$ | $53.16_{\pm 2.11}$ | $52.99_{\pm 2.18}$ | $52.11_{\pm 2.95}$ | $59.11_{\pm 1.88}$ |
| | | Mixing-up | $63.91_{\pm 2.39}$ | $64.29_{\pm 2.57}$ | $63.91_{\pm 2.39}$ | $63.70_{\pm 2.30}$ | $69.61_{\pm 3.64}$ | $68.89_{\pm 4.07}$ |
| | | TS-TCC | $49.14_{\pm 1.88}$ | $51.10_{\pm 8.54}$ | $49.14_{\pm 1.88}$ | $40.82_{\pm 4.17}$ | $48.05_{\pm 4.28}$ | $50.17_{\pm 3.94}$ |
| | | SimCLR | $59.46_{\pm 1.79}$ | $59.77_{\pm 1.87}$ | $59.46_{\pm 1.79}$ | $59.16_{\pm 1.81}$ | $59.77_{\pm 1.87}$ | $55.69_{\pm 1.27}$ |
| | | CLOCS | $58.85_{\pm 4.93}$ | $59.03_{\pm 5.00}$ | $58.85_{\pm 4.93}$ | $58.38_{\pm 5.45}$ | $62.57_{\pm 6.38}$ | $62.14_{\pm 6.34}$ |
| | | **COMET (Ours)** | $\mathbf{72.93_{\pm 7.21}}$ | $\mathbf{74.20_{\pm 7.68}}$ | $\mathbf{72.93_{\pm 7.21}}$ | $\mathbf{72.57_{\pm 7.37}}$ | $\mathbf{78.72_{\pm 8.42}}$ | $\mathbf{77.71_{\pm 9.10}}$ |

As a result, COMET demonstrates its superiority and stability across all the datasets. Furthermore, COMET outperforms SOTAs methods significantly with 10% and 1% label fractions, highlighting the effectiveness of our contrastive pre-training approach in reducing the reliance on labeled data.

## 5.3 Ablation Study, Visualization, Additional Downstream Tasks, and Heavy Duty Baseline

**Ablation study.** We verify the effectiveness of each contrastive block and other COMET variants. Besides, we study the impact of hyperparameter $\lambda$. (Appendix G.1)

**Visualization.** We visualize the embedding space and show our model can learn more distinctive and robust representations. (Appendix G.2)

**Additional downstream tasks.** Apart from the classification tasks in Section 5.1-5.2, we show the proposed COMET outperforms baselines in a wide range of downstream tasks, including clustering and anomaly detection. (Appendix G.3)

**Heavy duty baseline.** We show the superiority of our model does NOT result from the newly added contrastive blocks (*i.e.*, increased model parameters). COMET outperforms the heavy-duty SimCLR and TS2Vec with four contrastive blocks and four times pre-training epochs. (Appendix G.4)

**Table 4: Full fine-tuning results of ECG datasets**. A two-layer fully connected networks classifier $P$ is trained on top of pre-trained encoder $G$ on ECG dataset PTB. Fine-tuning the whole network includes $G$ and $P$. We use 100%, 10%, and 1% of labeled training data in PTB.

| Datasets | Fraction | Models | Accuracy | Precision | Recall | F1 score | AUROC | AUPRC |
|---|---|---|---|---|---|---|---|---|
| PTB | 100% | TS2vec | $85.14_{\pm1.66}$ | $87.82_{\pm2.21}$ | $76.84_{\pm3.99}$ | $79.66_{\pm3.63}$ | $90.50_{\pm1.59}$ | $\mathbf{90.07}_{\pm\mathbf{1.73}}$ |
| | | TF-C | $87.50_{\pm2.43}$ | $85.50_{\pm3.04}$ | $\mathbf{82.68}_{\pm\mathbf{4.04}}$ | $\mathbf{83.77}_{\pm\mathbf{3.50}}$ | $77.59_{\pm19.22}$ | $80.62_{\pm15.10}$ |
| | | Mixing-up | $87.61_{\pm1.48}$ | $\mathbf{89.56}_{\pm\mathbf{2.80}}$ | $79.30_{\pm1.67}$ | $82.56_{\pm2.00}$ | $89.29_{\pm1.26}$ | $88.94_{\pm1.04}$ |
| | | TS-TCC | $82.24_{\pm3.55}$ | $85.28_{\pm5.08}$ | $69.46_{\pm5.85}$ | $72.08_{\pm6.85}$ | $87.60_{\pm2.51}$ | $86.26_{\pm3.00}$ |
| | | SimCLR | $84.19_{\pm1.32}$ | $85.85_{\pm1.89}$ | $73.89_{\pm2.95}$ | $76.84_{\pm2.80}$ | $85.85_{\pm1.89}$ | $70.70_{\pm2.36}$ |
| | | CLOCS | $86.39_{\pm2.76}$ | $87.06_{\pm2.81}$ | $77.95_{\pm4.79}$ | $80.71_{\pm4.78}$ | $90.41_{\pm2.07}$ | $87.35_{\pm3.36}$ |
| | | **COMET (Ours)** | $\mathbf{87.84}_{\pm\mathbf{1.98}}$ | $87.67_{\pm1.72}$ | $81.14_{\pm3.68}$ | $83.45_{\pm3.22}$ | $\mathbf{92.95}_{\pm\mathbf{1.56}}$ | $87.47_{\pm2.82}$ |
| | 10% | TS2vec | $82.49_{\pm4.71}$ | $80.39_{\pm5.04}$ | $\mathbf{83.35}_{\pm\mathbf{0.91}}$ | $80.18_{\pm4.04}$ | $93.03_{\pm1.03}$ | $\mathbf{92.81}_{\pm\mathbf{0.97}}$ |
| | | TF-C | $85.37_{\pm1.23}$ | $82.80_{\pm2.35}$ | $79.94_{\pm0.71}$ | $81.09_{\pm1.14}$ | $81.57_{\pm15.60}$ | $84.57_{\pm8.12}$ |
| | | Mixing-up | $87.05_{\pm1.41}$ | $86.56_{\pm3.24}$ | $80.61_{\pm2.68}$ | $82.62_{\pm1.99}$ | $89.28_{\pm1.43}$ | $87.22_{\pm2.76}$ |
| | | TS-TCC | $83.38_{\pm1.53}$ | $85.11_{\pm2.11}$ | $72.24_{\pm2.45}$ | $75.27_{\pm2.64}$ | $86.06_{\pm1.76}$ | $84.34_{\pm2.08}$ |
| | | SimCLR | $83.84_{\pm2.15}$ | $87.19_{\pm1.34}$ | $72.51_{\pm4.63}$ | $75.44_{\pm4.77}$ | $87.19_{\pm1.34}$ | $69.99_{\pm3.84}$ |
| | | CLOCS | $88.25_{\pm2.48}$ | $88.64_{\pm2.12}$ | $81.40_{\pm4.64}$ | $83.84_{\pm4.03}$ | $91.91_{\pm2.40}$ | $89.76_{\pm3.94}$ |
| | | **COMET (Ours)** | $\mathbf{88.49}_{\pm\mathbf{3.28}}$ | $\mathbf{88.98}_{\pm\mathbf{2.60}}$ | $81.65_{\pm6.00}$ | $\mathbf{84.01}_{\pm\mathbf{5.61}}$ | $\mathbf{94.83}_{\pm\mathbf{1.08}}$ | $92.48_{\pm2.22}$ |
| | 1% | TS2vec | $81.95_{\pm1.85}$ | $78.78_{\pm0.98}$ | $83.83_{\pm0.36}$ | $79.77_{\pm1.44}$ | $90.99_{\pm1.34}$ | $88.69_{\pm1.44}$ |
| | | TF-C | $85.82_{\pm2.34}$ | $82.79_{\pm3.20}$ | $81.86_{\pm2.36}$ | $82.21_{\pm2.62}$ | $89.14_{\pm2.89}$ | $89.47_{\pm2.92}$ |
| | | Mixing-up | $84.71_{\pm4.11}$ | $82.33_{\pm6.07}$ | $78.17_{\pm4.89}$ | $79.81_{\pm5.33}$ | $89.04_{\pm3.57}$ | $84.67_{\pm5.16}$ |
| | | TS-TCC | $78.61_{\pm1.19}$ | $78.27_{\pm1.79}$ | $64.56_{\pm2.53}$ | $66.23_{\pm3.16}$ | $79.97_{\pm2.17}$ | $76.98_{\pm2.00}$ |
| | | SimCLR | $84.19_{\pm1.49}$ | $87.15_{\pm1.27}$ | $73.21_{\pm3.43}$ | $76.31_{\pm3.26}$ | $87.15_{\pm1.27}$ | $70.58_{\pm2.71}$ |
| | | CLOCS | $88.80_{\pm2.82}$ | $88.11_{\pm4.37}$ | $84.47_{\pm3.41}$ | $85.57_{\pm3.41}$ | $94.73_{\pm1.59}$ | $\mathbf{94.04}_{\pm\mathbf{2.14}}$ |
| | | **COMET (Ours)** | $\mathbf{90.52}_{\pm\mathbf{1.90}}$ | $\mathbf{88.58}_{\pm\mathbf{2.93}}$ | $\mathbf{88.23}_{\pm\mathbf{1.98}}$ | $\mathbf{88.23}_{\pm\mathbf{2.10}}$ | $\mathbf{95.08}_{\pm\mathbf{1.50}}$ | $93.78_{\pm1.98}$ |

# 6   Conclusion

This paper introduces COMET, a hierarchical contrastive representation learning framework tailored for medical time series. COMET leverages all data levels of medical time series, including patient, trial, sample, and observation levels, to capture the intricate complexities of the data. Through extensive experiments on three diverse datasets, we demonstrate that our method surpasses existing state-of-the-art methods in medical time series analysis. Our framework also shows its effectiveness in patient-independent downstream tasks, highlighting its potential to advance medical time series analysis and improve patient care and diagnosis.

One limitation of our work is the presence of label conflicts between different levels. In patient-level consistency, we assume all samples belonging to the same patient are positive while others are negative. In trial-level consistency, we consider samples from the same trial as positive samples and others as negative. This means a positive pair at the patient level may be considered negative at the trial level, as they do not belong to the same trial.

In future research, we aim to investigate the efficacy of our method across a wider range of datasets, with a particular focus on ECG datasets. Additionally, we intend to explore approaches to integrate disease-level consistency within our self-supervised contrastive framework.

We discuss the **broader impacts** with potential negative social impacts in Appendix H.

## Acknowledgments and Disclosure of Funding

This work is partially supported by the National Science Foundation under Grant No. 2245894. Any opinions, findings, conclusions or recommendations expressed in this material are those of the authors and do not necessarily reflect the views of the funders.

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

## Appendix A    A Medical Time Series Example

Here we use EEG data of Alzheimer's as an example. An EEG dataset has many patients with or without Alzheimer's (healthy control). Data collectors take multiple trials on a specific patient. These trials could be collected continuously within a short time or across a long period but follow the same experiment manner. Usually, the timestamps are too long for the deep learning pipeline to learn. For example, a 5 minutes trial with a sampling rate 256Hz has 76800 timestamps. Researchers generally use data preprocessing to split a trial into many samples, such as 1-second and 3-second short samples, for further representation learning. Each observation denotes a scalar or a vector of real value at a specific timestamp. An experiment with a sampling rate 256Hz has 256 observations in one second.

## Appendix B    Data Augmentation Banks

**Binomial masking**: Generate a mask following a binomial distribution that masks some timestamps of a sample, setting all channels at the masked timestamps to zero.

**Channel binomial masking**: Generate a mask following a binomial distribution that masks some channels of some timestamps of a sample, setting only a subset of channels at the masked timestamps to zero.

**Continuous masking**: Mask some continuous sequences of timestamps of a sample, setting all channels at the masked timestamps to zero.

**Channel continuous masking**: Mask some continuous sequences of timestamps of a sample, setting only a random half of the channels at the masked timestamps to zero.

**All true**: Do not apply any masking to the sample. Output the raw sample.

## Appendix C    Shuffle Function Banks

In real-world scenarios, ensuring the presence of samples from the same trial or patient within a training batch becomes increasingly low probability as the number of patients grows. This situation can hinder learning meaningful representations at the trial and patient levels. To address this situation, we have designed two distinct shuffle functions that serve to rearrange samples while also upholding the requirement to include samples originating from the same trial and patient. These functions are called the "trial shuffle" and the "batch shuffle".

**Trial shuffle**: This function shuffles samples originating from the same trial and subsequently shuffles the trial order. Initially, we arrange the samples by sorting them based on their trial IDs. Next, samples from the same trial are grouped into sets, and the order of samples within each trial set is shuffled. Finally, we sort the trial sets themselves while preserving the order of samples within each respective trial set.

**Batch shuffle**: This function shuffles samples in a batch and subsequently shuffles the order of batches. The logic of trial and batch shuffle are similar. Initially, we arrange the samples by sorting them based on their trial IDs. Next, we group samples into batch sets, and the order of samples within each batch set is shuffled. Finally, we sort the batch sets themselves while preserving the order of samples within each respective batch set.

**Random shuffle**: Besides the two specifically designed shuffle functions, this random shuffle function shuffles all the samples in the dataset.

All the shuffle functions mentioned here are designed to shuffle samples within the dataset before training. During training, it is also essential to shuffle samples each epoch to prevent the model from memorizing the dataset and encourage it to learn more useful representations. To address this, we implemented a specially crafted BatchSampler class in PyTorch, following the "batch shuffle" approach. This BatchSampler shuffles the samples locally within each epoch, ensuring that the pre-shuffled sample order is not disrupted significantly. This approach guarantees that each batch contains samples from the same trial. It's worth noting that when a batch consists of samples from the same trial, it must have samples from the same patient.

# Appendix D   Data Preprocessing

## D.1   AD Data Preprocessing

The AD dataset [44] comprises EEG recordings from 12 patients with Alzheimer's disease and 11 healthy controls. Each patient has an average of $30.0 \pm 12.5$ trials. Each trial corresponds to a 5-second interval with 1280 timestamps (sampled at 256Hz) and includes 16 channels. Prior to further processing, each trial is scaled using a standard scaler. To facilitate analysis, we segment each trial into nine half-overlapping samples, where each sample has a duration of 256 timestamps (equivalent to 1 second). Additionally, we assign a unique trial ID and patient ID to each sample based on its origin in terms of the patient and trial. We split training, validation, and test sets in a patient-independent way. We use samples from patient IDs 17 and 18 as the validation set and samples from IDs 19 and 20 as the test set. The rest of the samples are all put into the training set.

## D.2   PTB Data Preprocessing

The PTB dataset [45] consists of ECG recordings from 290 patients, with 15 channels sampled at 1000 Hz. There are a total of 8 types of heart diseases present in the dataset. For this paper, we focus on binary classification using a subset of the dataset that includes 198 patients with major disease labels, specifically Myocardial infarction and healthy controls. To preprocess the ECG signals, they are first normalized using a standard scaler after being resampled to a frequency of 250 Hz. Due to special peak information in ECG signals, a regular sliding window segmentation approach may result in the loss of crucial information. To address this issue, a different segmentation strategy is employed. Instead of sliding windows, the raw trials are segmented into individual heartbeats, with each heartbeat treated as a sample. To perform this segmentation, (1) the first step involves determining the maximum duration. The median value of R-Peak intervals across all channels is calculated for each raw trial, and outliers are removed to obtain a reasonable maximum interval as the standard heartbeat duration. (2) The next step is to determine the position of the first R-Peak. The median value of the first R-Peak position is calculated and used for all channels. (3) Next, the raw trials are split into individual heartbeat segments based on the median value of their respective R-Peak intervals. Each heartbeat is sampled starting from the R-Peak position, with the segments extending to both ends with half the length of the median interval. (4) To ensure the same length of the heartbeat samples, zero padding is applied to align their lengths with the maximum duration. (5) Finally, the samples are merged into trials, where 10 nearby samples are grouped together to form a pseudo-trial, similar to the neighborhood idea presented in [17]. We split training, validation, and test sets in a patient-independent way. We use samples from 28 patients(7 healthy and 21 positive) as the validation set and samples from another 28 patients(7 healthy and 21 positive) as the test set. The rest of the samples are all put into the training set.

## D.3   TDBrain Data Preprocessing

The TDBrain [46] is a large dataset that monitors the brain signals of 1274 patients with 33 channels (500 Hz) during EC (Eye closed) and EO (Eye open) tasks. The dataset consists of 60 types of diseases, and it is possible for a patient to have multiple diseases simultaneously. This paper focuses on a subset of the dataset, specifically 25 patients with Parkinson's disease and 25 healthy controls. Only the EC task trials are used for representation learning. To process the raw EC trials, we employ a sliding window approach that continuously moves from the middle of the trial to both ends without any overlap. Each raw EC trial is divided into processed pseudo-trials with a length of 2560 timestamps (10 seconds) after resampling to 256 Hz. These processed pseudo-trials are then scaled using a standard scaler. Furthermore, each pseudo-trial is split into 19 half-overlapping samples, with each sample having a length of 256 timestamps (1 second). In addition to the binary label indicating Parkinson's disease or healthy, each sample is assigned a patient and trial ID based on the patient and processed trial it originates from. It is important to note that the trial ID refers to the ID of the processed pseudo-trial and not the raw EC trial. We split training, validation, and test sets in a patient-independent way. We use samples from 8 patients(4 healthy and 4 positive) as the validation set and samples from another 8 patients(4 healthy and 4 positive) as the test set. The rest of the samples are all put into the training set.

# Appendix E  COMET and Baseline Implementation Details

We implement the baselines following their corresponding papers, including TS2vec [13], Mixing-up [16], TS-TCC [27], SimCLR [43], CLOCS [15], and TF-C [26]. In our COMET framework and all baselines, we use the last epoch of the contrastive pre-training encoder $G$ for downstream tasks. For the P-FT and F-FT tasks, we save the best model in terms of F1 score on the validation set during training and load the saved model to evaluate the performance on the test set.

**COMET (our model)** We use a two-layer, fully connected network as the projection head to map the input dimension to the output dimension. The input dimension corresponds to the feature dimension of the data, while the hidden dimension is set to 128, and the output dimension is set to 64. To augment the data, we apply time series masking on the output dimension using the [all_true, all_true, continuous, continuous] (see Appendix B) for the observation, sample, trial, and patient-level contrastive blocks, respectively. The augmented output dimension from the projection head is then passed to the encoder $G$ for representation learning. For the encoder $G$, we adopt a dilated CNN module. It consists of 10 hidden blocks, each following the order "GELU -> DilatedConv -> GELU -> DilatedConv." A residual connection is applied between the beginning and end of each block. The dilation factor of the convolution in the i-th block is set to $2^i$. Each hidden dimension of the dilated convolution is set to 64, and the kernel size is set to 3. The output dimension of encoder $G$ is fixed at 320. We utilize positive pair selection strategies specific to each contrastive block to build the contrastive loss in the embedding space (after encoder $G$). A max-pooling layer is employed before applying the representation to downstream tasks. During contrastive pre-training, we set the learning rate to 0.0001. The pre-training batch size is 256, and the total number of pre-training epochs is 100. We report the hyperparameters that achieved good results and stability among random seeds. The hyperparameters $\lambda_1$, $\lambda_2$, $\lambda_3$, $\lambda_4$ are assigned values of (0.25, 0.25, 0.25, 0.25), (0.1, 0.7, 0.1, 0.1), and (0.25, 0.25, 0.25, 0.25) for the AD, PTB, and TDBrain datasets, respectively.

**TS2vec** [13] introduces contextual consistency using overlapping subseries and a hierarchical loss function to capture data consistency at the observation and sample levels. To incorporate their methodology, we utilize the open-source code available at https://github.com/yuezhihan/ts2vec. However, their code does not include downstream tasks for F-FT (Full Fine-Tuning), so we implement these tasks using the same setup as our COMET. Specifically, we set the number of epochs for contrastive pre-training to 100, the learning rate to 0.0001, and the batch size to 256. To align with our model, we adjust the convolution blocks to 10, matching our configuration. We adopt the default settings provided by the TS2vec implementation for other settings during pre-training.

**TS-TCC** [27] leverages temporal and contextual consistency by contrasting a strong and weak augmentation. They employ a transformer-based autoregressive as the encoder. They perform a cross-view prediction by using temporal context to predict one view's future. Then, they maximize the similarity of contexts generated by the encoder to leverage contextual consistency. We utilize the open-source code available https://github.com/emadeldeen24/TS-TCC. We set the number of pre-training epochs to 100. We adopt the default settings provided by the TS-TCC implementation for other settings during pre-training.

**Mixing-up** [16] proposes a mixing-up augmentation by mixing the proportion of two time-series samples. This augmentation involves creating an augmented sample as the convex combination of two randomly selected time-series samples from the dataset. The mixing parameter follows a beta distribution, determining the proportion of the two samples in the augmentation process. We utilize the open-source code available at https://github.com/wickstrom/mixupcontrastivelearning to implement Mixing-up. Although they only provide downstream tasks for F-FT, we align their method with our setups for all downstream tasks, including P-FT and F-FT. We set the number of pre-training epochs to 100, the learning rate to 0.0001, and the batch size to 256 during pre-training.

**SimCLR** [18] is the most classic contrastive learning framework first proposed in the CV domain. It applies data augmentation techniques to create augmented views of input samples and constructs a contrastive loss based on these views. While initially designed for images, SimCLR has also been successfully adapted to time series data, as demonstrated in previous work such as [43]. To implement SimCLR, we utilize their open-source code available at https://github.com/iantangc/ContrastiveLearningHAR. We set the number of pre-training epochs to 100, the learning rate to 0.0001, and the batch size to 512, aligning with their recommended settings. We use the default values provided in the SimCLR implementation for other configuration parameters during pre-training.

**CLOCS** [15] employs samples from the same patient as positive pairs to leverage the data invariance in ECG recordings. They make use of both temporal and spatial information for contrastive learning. To implement CLOCS, we utilized their open-source code, available at https://github.com/danikiyasseh/CLOCS. We incorporated their contrastive loss function to implement the Contrastive Multi-segment Coding (CMSC) mechanism described in their paper. Additionally, we modified their backbone to use TCN, which shares the same network structure as our COMET, including the configuration parameters. Specifically, we set the number of pre-training epochs to 100, the learning rate to 0.0001, and the batch size to 256.

**TF-C** [26] leverages the consistency between time domain and frequency domain. They assume that the time-based and frequency-based representations of the same example exhibit proximity in the time-frequency space. We utilize their open-source code available at https://github.com/mims-harvard/TFC-pretraining to implement TF-C. While the original method is primarily designed for transfer learning, we extend it to incorporate downstream tasks such as P-FT and F-FT in our experiments. We set the number of pre-training epochs to 100, the learning rate to 0.0001, and the batch size to 256. We use the default values provided in the TF-C implementation for other configuration parameters during pre-training.

### E.1 Partial Fine-tuning

In the P-FT (Partial Fine-Tuning) setup, we introduce a classifier $L$ on top of the pre-trained encoder $G$, while keeping the parameters of $G$ fixed. Only the classifier $L$ is fine-tuned in this setup.

In the COMET, TS2vec, and Mixing-up approaches, we utilize logistic regression from the Sklearn library to implement the classifier $L$. We use the default settings of Sklearn, except for setting the maximum iteration to 100,000. We employ a one-layer fully connected network as the classifier $L$ for the TF-C method. The learning rates are specifically set to 8e-5, 3e-5, and 1e-4 for the AD, PTB, and TDBrain datasets, respectively. As for TS-TCC and SimCLR, we follow their respective default settings for the partial fine-tuning phase.

### E.2 Full Fine-tuning

In the F-FT (Full Fine-Tuning) setup, we introduce a classifier $P$ on top of the pre-trained encoder $G$, where both the parameters of the encoder $G$ and the classifier $P$ are trainable. In this setup, we fine-tune both the classifier $P$ and the encoder $G$. We utilize a fraction of 100%, 10%, and 1% labeled training data for fine-tuning.

In the COMET, TS2vec, and Mixing-up approaches, we set the finetune learning rate to 1e-4. We use a batch size of 128 and perform fine-tuning for 50, 100, and 100 epochs for 100%, 10%, and 1% label fractions, respectively. The classifier $P$ is implemented as a two-layer fully connected network with hidden dimensions 128. For TF-C, we change the hidden dimension of this two-layer fully connected network to 64. The learning rates are specifically set to 8e-5, 3e-5, and 1e-4 for the AD, PTB, and TDBrain datasets, respectively. Regarding TS-TCC and SimCLR, we follow their respective default settings for the full fine-tuning phase.

## Appendix F  Experimental Setting

### F.1 Patient-Independent Experimental Setting

This paper adopts a patient-independent setting for the train-validation-test split. In medical time series classification tasks, two common approaches to splitting the data are patient-independent and patient-dependent settings [15, S1]. Figure 3 illustrates these two settings' differences. In the patient-dependent setting, samples from the same patient can appear in both the training and testing sets, whereas in the patient-independent setting, samples from the same patient are exclusively included in either the training or testing set.

Performing patient-independent representation learning poses challenges due to the unique characteristics and different data distributions exhibited by each patient [50]. Even if two patients share the same label, the patterns within their data may differ significantly due to individual noise characteristics, potentially overshadowing the common patterns observed across patients. However, in real-world scenarios, it is essential for a model to be robust and general to perform patient-independent

representation learning. The goal is to train a model on a subset of patients with known labels and utilize it to predict other patients with unknown labels. In contrast, patient-dependent classification is usually impractical since it requires knowledge of the labels for all patients.

### F.2  Pseudo-Trial Experimental Setting

For many medical time series datasets, patient information is readily available, but trial information may be absent or limited to a single trial per patient. In such cases, the question arises of effectively employing trial-level contrastive learning. The solution is straightforward. We can generate pseudo-trials rather than merely deactivating the trial-level block by setting $\lambda$ to 0.

We can define groups of adjacent samples as pseudo-trials and assign them the same trial ID. In our paper, we employed ten neighboring samples as a pseudo-trial for the PTB and TDBrain datasets, and this approach yielded favorable results. This concept is akin to the approach taken by TNC [17], where close samples are defined as positive pairs.

## Appendix G   Ablation Study, Visualization, Additional Downstream Tasks, and Heavy Duty Baseline

### G.1  Ablation Study

**Ablation study of contrastive blocks**    We examined the effectiveness of each contrastive block and progressively incorporated each block from the observation-level to all four levels. Table 5 compares the full-level COMET model with its six variants on the AD, PTB, and TDBrain datasets. The variants are as follows: (1) **O**, **S**, **R**, **P**, which utilize only one level of the contrastive block. We activate that specific level by setting its $\lambda$ to 1 and deactivate the other three levels by setting their $\lambda$ to 0; (2) **S+O**, which combines the sample and observation-level contrastive blocks. The $\lambda$ values for sample and observation levels are set equally to 0.5; (3) **R+S+O**, incorporating the trial, sample, and observation-level contrastive blocks. The $\lambda$ values for these three levels are set equally to 0.33; and (4) **P+R+S+O**, representing our complete COMET method with all four levels of contrastive blocks. The $\lambda$ values for the four levels are set equally to 0.25. The "Fraction" column indicates the fraction of labeled training samples used during fine-tuning.

We observed that variants **R** and **P** exhibit strong performance, achieving comparable or even outperforming the full-level COMET model **P+R+S+O** on the PTB and TDBrain datasets across different fractions of labeled data (100%, 10%, and 1%). Notably, they also perform well with a 1% fraction in the AD dataset, although they exhibit significant instability in this case. This finding suggests that adding more contrastive blocks may not necessarily improve final results. Achieving a balance in weights among different levels through $\lambda$ is crucial. Nevertheless, training models at individual levels can provide insights into the importance of each level in contrastive representation learning. Furthermore, it's noteworthy that the full-level COMET model **P+R+S+O** consistently demonstrates comparable or superior results across all datasets and fractions. Importantly, we did not perform any parameter tuning here, opting to set all the $\lambda$ values equally, highlighting the stability of the full-level COMET **P+R+S+O** compared to other variants.

**Ablation study of hyperparameter $\lambda$**    We conducted an analysis to assess the impact of hyperparameter $\lambda$ on the AD, PTB, and TDBrain datasets in table 6. The values of $\lambda_1, \lambda_2, \lambda_3$, and $\lambda_4$, from left to right, correspond to the patient, trial, sample, and observation levels, respectively. In this analysis, all four data levels are incorporated, representing the full-level COMET model **P+R+S+O**. We applied a significant weight to one data level by setting its corresponding $\lambda$ to 0.7 while assigning lower weights of 0.1 to the other levels. Furthermore, we explored scenarios with increased weights on patient and trial levels or sample and observation levels. The "Fraction" column indicates the fraction of labeled training samples used during fine-tuning.

We observed that the results exhibited greater stability than the ablation study of contrastive blocks. In the contrastive block ablation study, there were significant discrepancies between different COMET variants at times. For instance, the **S** and **R** variants of the TDBrain dataset exhibited substantial differences across various fraction setups. However, in the full-level COMET model, the differences between these inter-running setups were notably reduced, even when a heavy weight was applied to one data level. For instance, the differences between lambda setups (0.1, 0.1, 0.7, 0.1) and (0.1,

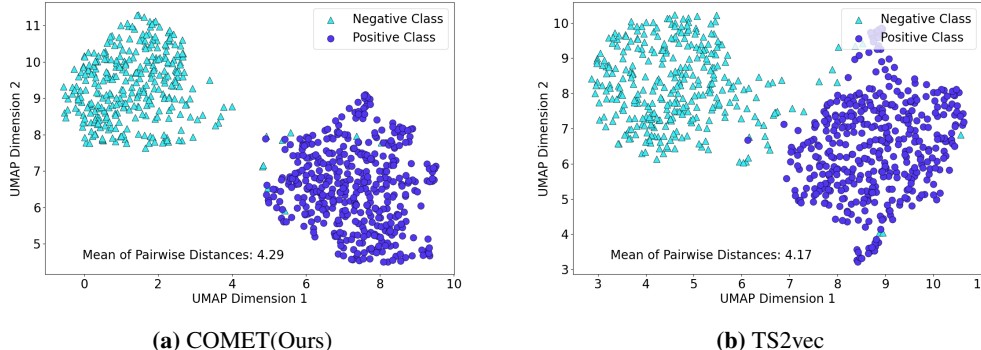

**(a)** COMET(Ours)          **(b)** TS2vec

**Figure 4: Visualizing the learned representation** (a) Visualization for TS2vec. (b) Visualization for COMET(Ours). The visualized representation is trained in the F-FT setup on the AD dataset. Dark blue and light blue denotes the negative class(Health) and positive class(Alzheimer), respectively. We calculate the mean Euclidean distance between pairwise samples from two classes for each pair of samples to evaluate the class separability. As the figure shows, our method exhibits superior separation between the two classes, resulting in larger pairwise distances.

0.7, 0.1, 0.1) were not as significant in the TDBrain dataset. This observation underscores again the stability and robustness of the full-level COMET model.

## G.2 Visualization

To visualize the effectiveness of COMET, we depict the learned representation $h_i$ using the F-FT setup on the AD dataset as a case study. It is important to note that the learned representation $h_i$ consists of 320 dimensions after pooling. To visualize the representations more interpretably, we employ UMAP, a dimensionality reduction technique with 20 neighbors and a minimum distance of 0.2. To establish a benchmark for comparison, we utilize TS2vec, which has shown the best performance among all baselines in the F-FT setup on the AD dataset. Additionally, we compute the average pairwise Euclidean distance between the negative (healthy) and positive (Alzheimer) classes, offering a quantitative measure of separability between the two classes.

## G.3 Performance on Downstream Tasks

**Clustering**  We assess the clustering performance of COMET using the AD dataset as a case study. Instead of employing a classifier model on top of the encoder, we apply K-means clustering (K=2) to the encoder $G$. We utilize three widely-used evaluation metrics: Silhouette score, Adjusted Rand Index (ARI), and Normalized Mutual Information (NMI). To establish a benchmark for comparison, we consider TS2vec, which has shown the best performance among all baselines in the F-FT setup on the AD dataset. Table 7 illustrates that COMET surpasses TS2vec with an improvement of 0.0586 in Silhouette score, 0.945 in ARI, and 0.881 in NMI.

**Anomaly detection**  We evaluate the anomaly detection performance of COMET using the AD dataset as a case study. While some previous works focus on identifying outlier observations within a sample [S2, S3], we concentrate on sample-level anomaly detection. We construct a very unbalanced AD test set comprising 90% negative (healthy) samples and 10% positive (Alzheimer's) samples. The negative samples are considered normal, while the positive samples are treated as outliers. The test set is prepared accordingly, while the remaining aspects of the experiment follow the F-FT setup. Specifically, We utilize the saved trained models from the F-FT setup to evaluate the new test set, and for comparison, we still employ TS2vec as a benchmark. The "Fraction" column indicates the fraction of labeled training samples used during fine-tuning. The experiment result is shown in table 8. The COMET outperforms TS2vec by 5.25%, 15.3% and 11.4% with label fraction 100%, 10% and 1%, respectively.

**Table 5: Ablation study of contrastive blocks**. The ablation study of contrastive blocks is evaluated on the AD, TDBrain, and PTB datasets. We examine the effectiveness of each contrastive block. Besides, we progressively incorporate each block from the observation-level to all four levels. Here, **O**, **S**, **R**, and **P** denote the observation, sample, trial, and patient-level contrastive blocks, respectively.

| Datasets | Fraction | Blocks | Accuracy | Precision | Recall | F1 score | AUROC | AUPRC |
|---|---|---|---|---|---|---|---|---|
| AD | 100% | O | 81.69±10.71 | 87.28±5.71 | 79.64±12.07 | 78.90±13.78 | 92.54±4.18 | 92.36±4.34 |
| | | S | 83.53±2.89 | 84.35±1.69 | 82.81±3.56 | 82.97±3.47 | 91.01±1.96 | 90.81±1.95 |
| | | R | 78.58±14.99 | 83.14±11.60 | 76.31±16.66 | 74.53±18.69 | 85.05±12.74 | 84.44±13.22 |
| | | P | 72.69±13.13 | 78.67±11.81 | 69.74±14.50 | 67.04±17.27 | 83.30±13.91 | 82.57±14.29 |
| | | S+O | **85.70±1.82** | 86.14±1.63 | **85.09±2.04** | **85.35±1.96** | 92.21±1.40 | 92.09±1.39 |
| | | R+S+O | 85.57±4.04 | 88.12±2.58 | 84.31±4.79 | 84.73±4.59 | 93.28±2.52 | 92.98±2.83 |
| | | P+R+S+O | 84.50±4.46 | **88.31±2.42** | 82.95±5.39 | 83.33±5.15 | **94.44±2.37** | **94.43±2.48** |
| | 10% | O | 86.35±9.25 | 87.09±9.04 | 85.55±9.92 | 85.70±10.10 | 92.62±8.99 | 92.77±8.72 |
| | | S | 77.30±3.63 | 78.45±3.75 | 76.26±3.91 | 76.36±3.92 | 85.43±3.61 | 84.63±3.76 |
| | | R | 77.83±17.59 | 83.45±14.35 | 75.41±19.62 | 72.08±23.45 | 83.51±16.90 | 83.27±17.00 |
| | | P | 71.41±15.31 | 76.91±15.56 | 68.70±16.46 | 66.54±17.74 | 76.11±16.98 | 76.06±16.19 |
| | | S+O | 82.73±2.05 | 84.33±2.71 | 81.51±2.07 | 81.98±2.11 | 90.02±2.47 | 90.02±2.40 |
| | | R+S+O | 91.19±3.14 | 91.74±3.01 | 90.70±3.34 | 90.98±3.25 | 95.86±2.63 | 95.86±2.67 |
| | | P+R+S+O | **91.43±3.12** | **92.52±2.36** | **90.71±3.56** | **91.14±3.31** | **96.44±2.84** | **96.48±2.82** |
| | 1% | O | 69.56±7.54 | 71.43±8.04 | 68.19±7.16 | 67.83±7.41 | 77.32±8.69 | 77.32±8.45 |
| | | S | 59.09±3.50 | 61.08±4.86 | 59.83±4.00 | 58.12±3.57 | 63.99±6.62 | 63.33±6.09 |
| | | R | **90.15±13.23** | **93.83±7.28** | **89.03±14.88** | **88.17±16.78** | **96.22±5.97** | 96.02±6.28 |
| | | P | 85.57±13.45 | 90.36±7.16 | 83.98±15.13 | 82.72±18.03 | 94.23±5.82 | 94.10±5.65 |
| | | S+O | 63.24±3.62 | 63.52±4.15 | 63.30±4.26 | 62.72±4.14 | 67.98±5.17 | 67.25±4.97 |
| | | R+S+O | 82.65±4.23 | 83.21±4.06 | 82.97±4.58 | 82.46±4.43 | 90.07±6.77 | 90.24±6.56 |
| | | P+R+S+O | 88.22±3.36 | 88.55±2.73 | 88.56±3.14 | 88.14±3.37 | 96.05±1.36 | **96.12±1.31** |
| PTB | 100% | O | 85.27±1.73 | 84.21±1.25 | 77.74±4.01 | 79.78±3.37 | 88.13±2.23 | 84.76±2.14 |
| | | S | 84.30±2.56 | 84.00±2.16 | 75.68±5.69 | 77.74±5.13 | 88.66±2.05 | 84.80±2.15 |
| | | R | 88.63±1.43 | 88.42±1.45 | 82.46±2.35 | 84.72±2.12 | 89.29±3.96 | 86.21±4.96 |
| | | P | **88.85±3.22** | **88.74±2.30** | **82.78±6.11** | **84.75±5.38** | **94.32±1.81** | **90.61±2.81** |
| | | S+O | 84.38±2.34 | 84.33±1.18 | 75.49±5.69 | 77.69±4.76 | 90.37±2.59 | 86.57±4.32 |
| | | R+S+O | 85.32±1.93 | 84.82±1.78 | 77.54±4.58 | 79.64±3.94 | 92.34±1.85 | 88.75±1.98 |
| | | P+R+S+O | 86.36±1.44 | 87.18±1.28 | 77.87±2.55 | 80.79±2.44 | 93.41±2.35 | 89.09±1.64 |
| | 10% | O | 86.75±1.44 | 85.42±2.10 | 80.88±3.27 | 82.45±2.29 | 90.71±3.16 | 88.84±3.59 |
| | | S | 86.34±0.73 | 84.75±1.49 | 80.21±1.78 | 81.90±1.25 | 88.36±0.31 | 85.20±1.17 |
| | | R | 88.12±1.75 | 86.36±2.28 | 84.16±3.55 | 84.80±2.39 | 91.80±2.50 | 90.45±1.90 |
| | | P | **90.38±2.23** | **90.33±2.71** | 85.69±4.36 | 87.27±3.45 | **93.06±3.86** | **91.83±2.93** |
| | | S+O | 85.84±1.74 | 84.14±0.76 | 80.07±5.18 | 81.14±3.82 | 90.28±2.75 | 87.57±3.42 |
| | | R+S+O | 85.88±3.08 | 83.60±3.61 | 82.45±1.94 | 82.48±2.71 | 90.79±1.79 | 88.68±1.91 |
| | | P+R+S+O | 90.32±1.61 | 89.67±1.94 | **85.85±2.99** | **87.35±2.36** | 92.78±1.11 | 90.46±3.09 |
| | 1% | O | 85.73±1.59 | 83.79±2.15 | 79.53±3.14 | 81.09±2.54 | 90.15±2.32 | 88.85±2.64 |
| | | S | 77.63±1.72 | 72.36±2.41 | 69.41±2.53 | 70.38±2.29 | 79.09±2.21 | 75.07±1.88 |
| | | R | 86.28±1.97 | 84.02±2.85 | 82.06±2.08 | 82.61±1.97 | 90.50±0.99 | 88.32±1.40 |
| | | P | **91.55±2.45** | **90.28±3.27** | **88.45±3.22** | **89.25±3.13** | **95.12±2.58** | **94.80±2.04** |
| | | S+O | 81.78±1.07 | 77.65±1.34 | 77.17±2.14 | 77.20±1.43 | 86.65±2.38 | 82.12±1.95 |
| | | R+S+O | 80.71±2.28 | 76.79±2.03 | 80.51±1.67 | 77.84±2.13 | 88.76±1.38 | 85.00±2.23 |
| | | P+R+S+O | 86.62±3.90 | 82.99±4.52 | 85.36±5.73 | 83.89±4.96 | 93.78±3.82 | 91.94±4.96 |
| TDBrain | 100% | O | 89.92±2.15 | 90.50±2.07 | 89.92±2.15 | 89.89±2.17 | 96.79±1.29 | 96.84±1.29 |
| | | S | 77.86±2.85 | 78.98±2.51 | 77.86±2.85 | 77.62±3.01 | 88.44±2.23 | 88.96±2.14 |
| | | R | **94.44±1.79** | **94.60±1.75** | **94.44±1.79** | **94.44±1.80** | **98.39±1.29** | **98.31±1.46** |
| | | P | 93.18±3.70 | 93.27±3.67 | 93.18±3.70 | 93.18±3.70 | 97.59±2.01 | 97.58±2.00 |
| | | S+O | 80.38±4.22 | 81.59±3.70 | 80.38±4.22 | 80.16±4.38 | 91.35±2.24 | 91.64±2.05 |
| | | R+S+O | 86.01±4.29 | 86.35±3.81 | 86.01±4.29 | 85.95±4.38 | 94.14±2.72 | 94.37±2.58 |
| | | P+R+S+O | 85.47±1.16 | 85.68±1.20 | 85.47±1.16 | 85.45±1.16 | 93.73±1.02 | 93.96±0.99 |
| | 10% | O | 85.34±4.38 | 86.03±3.97 | 85.34±4.38 | 85.25±4.48 | 93.18±3.52 | 93.24±3.53 |
| | | S | 74.02±2.09 | 74.62±2.01 | 74.02±2.09 | 73.86±2.17 | 81.92±3.25 | 81.76±3.42 |
| | | R | **92.96±8.22** | **93.02±8.20** | **92.96±8.22** | **92.96±8.23** | **96.14±5.80** | **95.93±6.08** |
| | | P | 89.38±14.69 | 89.58±14.72 | 89.38±14.69 | 89.36±14.69 | 93.23±11.81 | 93.52±11.10 |
| | | S+O | 74.92±2.57 | 76.60±3.07 | 74.92±2.57 | 74.54±2.55 | 84.51±3.07 | 84.40±3.07 |
| | | R+S+O | 81.90±4.74 | 83.55±4.02 | 81.90±4.74 | 81.61±4.99 | 91.21±3.52 | 90.73±3.72 |
| | | P+R+S+O | 79.28±4.64 | 79.83±4.83 | 79.28±4.64 | 79.19±4.62 | 88.39±4.13 | 88.38±3.96 |
| | 1% | O | 71.52±7.54 | 72.17±7.71 | 71.52±7.54 | 71.31±7.62 | 78.32±8.37 | 77.56±8.91 |
| | | S | 58.62±3.84 | 59.66±3.95 | 58.62±3.84 | 57.39±4.26 | 62.85±6.38 | 61.61±6.61 |
| | | R | **85.29±5.93** | **85.55±5.66** | **85.29±5.93** | **85.24±5.99** | **91.19±4.76** | **91.11±4.74** |
| | | P | 77.23±3.42 | 78.10±3.39 | 77.23±3.42 | 77.04±3.47 | 86.12±3.88 | 85.10±4.11 |
| | | S+O | 61.71±2.97 | 61.82±2.90 | 61.71±2.97 | 61.60±3.07 | 66.09±2.98 | 64.94±2.66 |
| | | R+S+O | 72.15±6.26 | 73.39±7.50 | 72.15±6.26 | 71.91±6.12 | 78.39±7.77 | 76.97±7.61 |
| | | P+R+S+O | 72.93±7.21 | 74.20±7.68 | 72.93±7.21 | 72.57±7.37 | 78.72±8.42 | 77.71±9.10 |

Table 6: **Ablation study of hyperparameter** $\lambda$. The ablation study of hyperparameter $\lambda$ is evaluated on the AD, TDBrain, and PTB datasets. The $\lambda_1, \lambda_2, \lambda_3, \lambda_4$ from left to right are for patient, trial, sample, and observation levels, respectively.

| Datasets | Fraction | $\lambda_1, \lambda_2, \lambda_3, \lambda_4$ | Accuracy | Precision | Recall | F1 score | AUROC | AUPRC |
|---|---|---|---|---|---|---|---|---|
| AD | 100% | (0.1,0.1,0.1,0.7) | $\mathbf{87.82_{\pm7.44}}$ | $\mathbf{91.08_{\pm4.10}}$ | $86.62_{\pm8.58}$ | $86.77_{\pm8.64}$ | $\mathbf{97.95_{\pm1.26}}$ | $\mathbf{97.94_{\pm1.26}}$ |
| | | (0.1,0.1,0.7,0.1) | $85.03_{\pm5.64}$ | $89.09_{\pm3.22}$ | $83.32_{\pm6.37}$ | $83.78_{\pm6.59}$ | $95.33_{\pm1.81}$ | $95.30_{\pm1.90}$ |
| | | (0.1,0.7,0.1,0.1) | $82.76_{\pm8.67}$ | $88.00_{\pm4.65}$ | $80.98_{\pm10.02}$ | $80.72_{\pm10.30}$ | $95.32_{\pm2.72}$ | $95.26_{\pm2.71}$ |
| | | (0.7,0.1,0.1,0.1) | $82.22_{\pm8.31}$ | $87.27_{\pm4.41}$ | $80.42_{\pm9.62}$ | $80.17_{\pm10.00}$ | $94.84_{\pm2.64}$ | $94.80_{\pm2.65}$ |
| | | (0.2,0.2,0.3,0.3) | $86.88_{\pm5.78}$ | $89.96_{\pm4.00}$ | $85.46_{\pm6.52}$ | $85.97_{\pm6.36}$ | $94.03_{\pm3.56}$ | $94.07_{\pm3.54}$ |
| | | (0.3,0.3,0.2,0.2) | $86.72_{\pm5.97}$ | $89.76_{\pm3.12}$ | $85.41_{\pm6.92}$ | $85.73_{\pm6.96}$ | $95.14_{\pm1.95}$ | $95.09_{\pm2.04}$ |
| | | (0.3,0.3,0.15,0.25) | $82.84_{\pm6.96}$ | $88.09_{\pm3.99}$ | $80.87_{\pm7.93}$ | $81.04_{\pm8.06}$ | $94.73_{\pm2.98}$ | $94.64_{\pm3.12}$ |
| | 10% | (0.1,0.1,0.1,0.7) | $89.56_{\pm8.67}$ | $91.37_{\pm6.73}$ | $88.64_{\pm9.65}$ | $88.82_{\pm9.51}$ | $96.55_{\pm4.41}$ | $96.47_{\pm4.63}$ |
| | | (0.1,0.1,0.7,0.1) | $87.84_{\pm6.09}$ | $88.72_{\pm5.16}$ | $87.50_{\pm6.76}$ | $87.42_{\pm6.66}$ | $94.58_{\pm5.32}$ | $94.38_{\pm5.67}$ |
| | | (0.1,0.7,0.1,0.1) | $79.89_{\pm14.06}$ | $85.79_{\pm7.50}$ | $77.97_{\pm16.05}$ | $75.54_{\pm19.81}$ | $92.89_{\pm6.45}$ | $92.67_{\pm6.64}$ |
| | | (0.7,0.1,0.1,0.1) | $78.26_{\pm13.58}$ | $84.99_{\pm6.68}$ | $76.38_{\pm15.80}$ | $73.44_{\pm19.24}$ | $92.74_{\pm7.64}$ | $92.57_{\pm7.93}$ |
| | | (0.2,0.2,0.3,0.3) | $\mathbf{93.25_{\pm1.71}}$ | $\mathbf{93.86_{\pm1.24}}$ | $\mathbf{92.77_{\pm2.00}}$ | $\mathbf{93.09_{\pm1.80}}$ | $\mathbf{97.68_{\pm0.87}}$ | $\mathbf{97.75_{\pm0.82}}$ |
| | | (0.3,0.3,0.2,0.2) | $91.43_{\pm3.70}$ | $92.71_{\pm2.39}$ | $90.68_{\pm4.28}$ | $91.09_{\pm4.00}$ | $96.81_{\pm1.62}$ | $96.88_{\pm1.64}$ |
| | | (0.3,0.3,0.15,0.25) | $91.54_{\pm3.63}$ | $92.93_{\pm2.32}$ | $90.77_{\pm4.25}$ | $91.19_{\pm3.91}$ | $97.26_{\pm1.63}$ | $97.29_{\pm1.67}$ |
| | 1% | (0.1,0.1,0.1,0.7) | $95.88_{\pm2.29}$ | $96.21_{\pm1.92}$ | $95.64_{\pm2.55}$ | $95.79_{\pm2.36}$ | $92.72_{\pm9.34}$ | $93.24_{\pm8.74}$ |
| | | (0.1,0.1,0.7,0.1) | $85.03_{\pm9.80}$ | $87.16_{\pm5.75}$ | $85.65_{\pm8.54}$ | $84.62_{\pm10.42}$ | $94.71_{\pm2.73}$ | $94.73_{\pm2.89}$ |
| | | (0.1,0.7,0.1,0.1) | $\mathbf{97.19_{\pm1.28}}$ | $\mathbf{97.32_{\pm1.11}}$ | $\mathbf{97.08_{\pm1.42}}$ | $\mathbf{97.15_{\pm1.31}}$ | $\mathbf{99.02_{\pm1.06}}$ | $\mathbf{98.98_{\pm1.11}}$ |
| | | (0.7,0.1,0.1,0.1) | $97.00_{\pm1.10}$ | $97.10_{\pm1.06}$ | $96.92_{\pm1.14}$ | $96.96_{\pm1.12}$ | $98.93_{\pm1.20}$ | $98.89_{\pm1.26}$ |
| | | (0.2,0.2,0.3,0.3) | $84.55_{\pm7.39}$ | $84.47_{\pm7.51}$ | $84.75_{\pm7.65}$ | $84.44_{\pm7.49}$ | $89.95_{\pm10.83}$ | $89.78_{\pm11.27}$ |
| | | (0.3,0.3,0.2,0.2) | $86.96_{\pm5.92}$ | $87.16_{\pm5.66}$ | $86.87_{\pm6.31}$ | $86.72_{\pm6.19}$ | $92.68_{\pm7.76}$ | $92.93_{\pm7.43}$ |
| | | (0.3,0.3,0.15,0.25) | $92.40_{\pm2.52}$ | $92.75_{\pm2.41}$ | $92.16_{\pm2.44}$ | $92.27_{\pm2.55}$ | $97.76_{\pm1.11}$ | $97.76_{\pm1.17}$ |
| PTB | 100% | (0.1,0.1,0.1,0.7) | $85.51_{\pm2.46}$ | $85.54_{\pm2.52}$ | $76.96_{\pm4.20}$ | $79.60_{\pm3.91}$ | $91.30_{\pm2.28}$ | $87.11_{\pm3.84}$ |
| | | (0.1,0.1,0.7,0.1) | $85.94_{\pm3.40}$ | $87.23_{\pm4.03}$ | $76.69_{\pm5.61}$ | $79.64_{\pm5.94}$ | $93.78_{\pm2.51}$ | $\mathbf{90.78_{\pm4.78}}$ |
| | | (0.1,0.7,0.1,0.1) | $\mathbf{87.84_{\pm1.98}}$ | $87.67_{\pm1.72}$ | $\mathbf{81.14_{\pm3.68}}$ | $\mathbf{83.45_{\pm3.22}}$ | $92.95_{\pm1.56}$ | $87.47_{\pm2.82}$ |
| | | (0.7,0.1,0.1,0.1) | $87.76_{\pm2.75}$ | $\mathbf{88.20_{\pm1.41}}$ | $80.65_{\pm5.54}$ | $82.96_{\pm5.07}$ | $91.82_{\pm3.66}$ | $88.70_{\pm3.08}$ |
| | | (0.2,0.2,0.3,0.3) | $87.74_{\pm1.46}$ | $87.78_{\pm1.48}$ | $80.79_{\pm2.58}$ | $83.28_{\pm2.31}$ | $92.97_{\pm1.97}$ | $88.67_{\pm1.71}$ |
| | | (0.3,0.3,0.2,0.2) | $87.16_{\pm2.02}$ | $87.62_{\pm1.60}$ | $79.47_{\pm3.69}$ | $82.15_{\pm3.33}$ | $\mathbf{94.00_{\pm0.42}}$ | $88.83_{\pm1.65}$ |
| | | (0.3,0.3,0.15,0.25) | $87.56_{\pm0.87}$ | $87.91_{\pm0.64}$ | $80.25_{\pm1.69}$ | $82.92_{\pm1.50}$ | $91.85_{\pm2.44}$ | $88.27_{\pm2.34}$ |
| | 10% | (0.1,0.1,0.1,0.7) | $87.76_{\pm2.62}$ | $87.15_{\pm2.27}$ | $81.53_{\pm5.10}$ | $83.43_{\pm4.15}$ | $92.54_{\pm2.17}$ | $89.85_{\pm2.65}$ |
| | | (0.1,0.1,0.7,0.1) | $88.29_{\pm2.53}$ | $85.80_{\pm3.57}$ | $\mathbf{86.87_{\pm1.39}}$ | $85.89_{\pm2.34}$ | $94.56_{\pm0.80}$ | $\mathbf{93.15_{\pm0.75}}$ |
| | | (0.1,0.7,0.1,0.1) | $88.49_{\pm3.28}$ | $88.98_{\pm2.60}$ | $81.65_{\pm6.00}$ | $84.01_{\pm5.61}$ | $\mathbf{94.83_{\pm1.08}}$ | $92.48_{\pm2.22}$ |
| | | (0.7,0.1,0.1,0.1) | $88.70_{\pm3.20}$ | $\mathbf{89.47_{\pm2.40}}$ | $81.85_{\pm6.02}$ | $84.26_{\pm5.45}$ | $94.30_{\pm1.76}$ | $93.24_{\pm2.41}$ |
| | | (0.2,0.2,0.3,0.3) | $89.25_{\pm2.07}$ | $88.87_{\pm2.55}$ | $83.98_{\pm3.97}$ | $85.72_{\pm3.13}$ | $94.26_{\pm1.47}$ | $90.82_{\pm3.43}$ |
| | | (0.3,0.3,0.2,0.2) | $89.39_{\pm2.02}$ | $88.86_{\pm2.90}$ | $84.43_{\pm3.65}$ | $86.03_{\pm3.00}$ | $93.59_{\pm1.13}$ | $92.42_{\pm2.00}$ |
| | | (0.3,0.3,0.15,0.25) | $\mathbf{89.82_{\pm3.00}}$ | $89.36_{\pm2.16}$ | $84.71_{\pm5.73}$ | $\mathbf{86.34_{\pm4.84}}$ | $93.55_{\pm1.96}$ | $90.54_{\pm3.20}$ |
| | 1% | (0.1,0.1,0.1,0.7) | $89.45_{\pm1.80}$ | $87.49_{\pm2.15}$ | $86.23_{\pm3.61}$ | $86.62_{\pm2.57}$ | $94.03_{\pm2.59}$ | $93.21_{\pm2.60}$ |
| | | (0.1,0.1,0.7,0.1) | $85.12_{\pm2.24}$ | $81.33_{\pm2.60}$ | $83.06_{\pm3.24}$ | $82.01_{\pm2.76}$ | $91.12_{\pm2.14}$ | $88.96_{\pm2.92}$ |
| | | (0.1,0.7,0.1,0.1) | $\mathbf{90.52_{\pm1.90}}$ | $\mathbf{88.58_{\pm2.93}}$ | $88.23_{\pm1.98}$ | $\mathbf{88.23_{\pm2.10}}$ | $95.08_{\pm1.50}$ | $93.78_{\pm1.98}$ |
| | | (0.7,0.1,0.1,0.1) | $90.19_{\pm1.75}$ | $87.88_{\pm2.23}$ | $88.20_{\pm2.70}$ | $87.87_{\pm2.19}$ | $94.82_{\pm1.65}$ | $93.68_{\pm2.24}$ |
| | | (0.2,0.2,0.3,0.3) | $85.28_{\pm4.39}$ | $81.48_{\pm5.03}$ | $84.10_{\pm5.66}$ | $82.47_{\pm5.28}$ | $92.87_{\pm3.85}$ | $90.34_{\pm4.96}$ |
| | | (0.3,0.3,0.2,0.2) | $86.96_{\pm2.56}$ | $83.38_{\pm2.89}$ | $85.65_{\pm3.59}$ | $84.32_{\pm3.15}$ | $94.47_{\pm1.90}$ | $92.31_{\pm2.67}$ |
| | | (0.3,0.3,0.15,0.25) | $89.43_{\pm1.38}$ | $86.42_{\pm1.96}$ | $\mathbf{88.86_{\pm0.94}}$ | $87.37_{\pm1.39}$ | $\mathbf{95.32_{\pm1.04}}$ | $\mathbf{94.29_{\pm1.72}}$ |
| TDBrain | 100% | (0.1,0.1,0.1,0.7) | $87.20_{\pm3.39}$ | $87.40_{\pm3.34}$ | $87.20_{\pm3.39}$ | $87.18_{\pm3.40}$ | $94.84_{\pm2.34}$ | $94.96_{\pm2.33}$ |
| | | (0.1,0.1,0.7,0.1) | $\mathbf{88.53_{\pm3.86}}$ | $\mathbf{88.66_{\pm3.79}}$ | $\mathbf{88.53_{\pm3.86}}$ | $\mathbf{88.52_{\pm3.87}}$ | $95.57_{\pm2.74}$ | $95.71_{\pm2.66}$ |
| | | (0.1,0.7,0.1,0.1) | $87.97_{\pm3.26}$ | $88.33_{\pm3.15}$ | $87.97_{\pm3.26}$ | $87.94_{\pm3.28}$ | $\mathbf{95.58_{\pm1.92}}$ | $\mathbf{95.77_{\pm1.86}}$ |
| | | (0.7,0.1,0.1,0.1) | $87.25_{\pm2.71}$ | $87.54_{\pm2.74}$ | $87.25_{\pm2.71}$ | $87.22_{\pm2.71}$ | $95.14_{\pm1.73}$ | $95.32_{\pm1.70}$ |
| | | (0.2,0.2,0.3,0.3) | $85.69_{\pm2.42}$ | $86.00_{\pm2.53}$ | $85.69_{\pm2.42}$ | $85.66_{\pm2.41}$ | $94.14_{\pm1.43}$ | $94.31_{\pm1.28}$ |
| | | (0.3,0.3,0.2,0.2) | $85.76_{\pm3.18}$ | $86.24_{\pm2.75}$ | $85.76_{\pm3.18}$ | $85.69_{\pm3.24}$ | $94.34_{\pm1.79}$ | $94.45_{\pm1.77}$ |
| | | (0.3,0.3,0.15,0.25) | $85.54_{\pm4.11}$ | $86.30_{\pm3.62}$ | $85.54_{\pm4.11}$ | $85.44_{\pm4.21}$ | $94.22_{\pm2.86}$ | $94.36_{\pm2.80}$ |
| | 10% | (0.1,0.1,0.1,0.7) | $76.92_{\pm4.84}$ | $78.35_{\pm3.64}$ | $76.92_{\pm4.84}$ | $76.50_{\pm5.33}$ | $85.85_{\pm3.99}$ | $85.36_{\pm4.30}$ |
| | | (0.1,0.1,0.7,0.1) | $81.91_{\pm5.84}$ | $82.40_{\pm5.82}$ | $81.91_{\pm5.84}$ | $81.83_{\pm5.87}$ | $90.38_{\pm5.11}$ | $90.72_{\pm5.16}$ |
| | | (0.1,0.7,0.1,0.1) | $84.51_{\pm4.81}$ | $84.90_{\pm4.61}$ | $84.51_{\pm4.81}$ | $84.45_{\pm4.86}$ | $92.58_{\pm3.30}$ | $92.62_{\pm3.09}$ |
| | | (0.7,0.1,0.1,0.1) | $\mathbf{84.92_{\pm5.83}}$ | $\mathbf{85.24_{\pm5.43}}$ | $\mathbf{84.92_{\pm5.83}}$ | $\mathbf{84.86_{\pm5.94}}$ | $\mathbf{92.69_{\pm4.07}}$ | $\mathbf{92.69_{\pm3.81}}$ |
| | | (0.2,0.2,0.3,0.3) | $80.79_{\pm3.84}$ | $82.00_{\pm3.76}$ | $80.79_{\pm3.84}$ | $80.59_{\pm3.94}$ | $89.88_{\pm3.50}$ | $89.56_{\pm3.76}$ |
| | | (0.3,0.3,0.2,0.2) | $78.93_{\pm3.88}$ | $79.78_{\pm4.06}$ | $78.93_{\pm3.88}$ | $78.77_{\pm3.94}$ | $88.64_{\pm3.52}$ | $88.59_{\pm3.24}$ |
| | | (0.3,0.3,0.15,0.25) | $79.96_{\pm5.63}$ | $81.00_{\pm5.52}$ | $79.96_{\pm5.63}$ | $79.76_{\pm5.76}$ | $88.89_{\pm4.57}$ | $88.70_{\pm4.21}$ |
| | 1% | (0.1,0.1,0.1,0.7) | $68.68_{\pm2.88}$ | $70.46_{\pm3.56}$ | $68.68_{\pm2.88}$ | $68.02_{\pm2.95}$ | $73.79_{\pm4.80}$ | $72.99_{\pm5.27}$ |
| | | (0.1,0.1,0.7,0.1) | $66.49_{\pm8.01}$ | $68.46_{\pm7.64}$ | $66.49_{\pm8.01}$ | $65.03_{\pm9.26}$ | $73.30_{\pm8.15}$ | $72.74_{\pm8.32}$ |
| | | (0.1,0.7,0.1,0.1) | $73.08_{\pm8.61}$ | $\mathbf{74.89_{\pm8.18}}$ | $73.08_{\pm8.61}$ | $72.45_{\pm8.89}$ | $78.39_{\pm4.58}$ | $77.79_{\pm4.18}$ |
| | | (0.7,0.1,0.1,0.1) | $73.16_{\pm8.72}$ | $75.14_{\pm8.13}$ | $73.16_{\pm8.72}$ | $72.46_{\pm9.05}$ | $78.42_{\pm4.48}$ | $77.76_{\pm4.05}$ |
| | | (0.2,0.2,0.3,0.3) | $71.55_{\pm8.49}$ | $72.14_{\pm9.14}$ | $71.55_{\pm8.49}$ | $71.44_{\pm8.44}$ | $77.14_{\pm9.24}$ | $76.04_{\pm9.79}$ |
| | | (0.3,0.3,0.2,0.2) | $70.11_{\pm8.07}$ | $71.68_{\pm8.95}$ | $70.11_{\pm8.07}$ | $69.54_{\pm8.38}$ | $76.24_{\pm10.55}$ | $75.26_{\pm10.84}$ |
| | | (0.3,0.3,0.15,0.25) | $\mathbf{73.59_{\pm9.28}}$ | $74.53_{\pm9.61}$ | $\mathbf{73.59_{\pm9.28}}$ | $\mathbf{73.33_{\pm9.36}}$ | $\mathbf{79.07_{\pm9.67}}$ | $\mathbf{77.97_{\pm9.62}}$ |

**Table 7: Performance on downstream clustering.** The clustering performance is evaluated on the AD dataset. We compare the baseline TS2vec, which performs best in the F-FT setup.

| Method | Silhouette | ARI | NMI |
|---|---|---|---|
| Random Init. | $0.1184 \pm 0.0082$ | $0.1189 \pm 0.0664$ | $0.1258 \pm 0.0660$ |
| TS2vec | $0.0795 \pm 0.0032$ | $0.0013 \pm 0.0026$ | $0.0018 \pm 0.0016$ |
| COMET (Ours) | $0.1381 \pm 0.0139$ | $0.9358 \pm 0.0264$ | $0.8827 \pm 0.0414$ |

**Table 8: Performance on anomaly detection**. Sample-level anomaly detection on a very unbalanced AD test set comprising 90% negative (healthy) samples and 10% positive (Alzheimer) samples.

| Fraction | Models | Accuracy | Precision | Recall | F1 score | AUROC | AUPRC |
|---|---|---|---|---|---|---|---|
| 100% | TS2vec | $82.11_{\pm 3.30}$ | $66.05_{\pm 2.45}$ | $82.13_{\pm 3.21}$ | $68.70_{\pm 3.37}$ | $91.27_{\pm 1.47}$ | $76.80_{\pm 3.08}$ |
| | COMET (Ours) | $83.03_{\pm 11.65}$ | $71.76_{\pm 7.60}$ | $90.33_{\pm 6.31}$ | $73.95_{\pm 11.97}$ | $97.99_{\pm 1.37}$ | $91.52_{\pm 5.52}$ |
| 10% | TS2vec | $76.05_{\pm 6.35}$ | $62.48_{\pm 2.90}$ | $77.33_{\pm 4.74}$ | $62.67_{\pm 5.00}$ | $86.76_{\pm 3.95}$ | $72.21_{\pm 5.09}$ |
| | COMET (Ours) | $88.22_{\pm 2.88}$ | $73.22_{\pm 3.31}$ | $92.49_{\pm 1.73}$ | $77.97_{\pm 3.87}$ | $97.91_{\pm 1.14}$ | $92.21_{\pm 4.43}$ |
| 1% | TS2vec | $67.24_{\pm 8.05}$ | $57.34_{\pm 3.12}$ | $67.87_{\pm 6.45}$ | $54.35_{\pm 6.21}$ | $72.75_{\pm 6.82}$ | $61.32_{\pm 5.25}$ |
| | COMET (Ours) | $77.57_{\pm 4.21}$ | $64.72_{\pm 1.95}$ | $84.41_{\pm 3.38}$ | $65.74_{\pm 3.67}$ | $93.13_{\pm 2.82}$ | $79.65_{\pm 7.65}$ |

### G.4 Heavy Duty Baselines

In COMET, we incorporate four contrastive blocks to leverage four levels of data consistency, allowing the data to pass through the model four times within one epoch during contrastive pre-training. To ensure that our superior performance is not due to increased data passing, we conduct experiments on the AD dataset with two baselines: SimCLR and TS2vec.

SimCLR utilizes only one contrastive block during training. We employ two strategies to match the data passing number with COMET: (1) Run SimCLR with one contrastive block for four times the original number of epochs in pre-training (400 epochs instead of 100). (2) Duplicate the contrastive blocks, resulting in four SimCLR contrastive blocks. The notation **4E** signifies running the model for four times the original number of epochs, while **4B** indicates the use of four times the number of contrastive blocks compared to the original SimCLR.

TS2vec incorporates two contrastive blocks during training, leading to data passing twice within one epoch. Similarly, we adopt two strategies to align the data passing number with COMET: (1) Run TS2vec for two times the original number of epochs in pre-training (200 epochs instead of 100). (2) Duplicate the contrastive blocks, resulting in four TS2vec contrastive blocks. The notation **2E** denotes running the model for four times the original number of epochs, while **2B** indicates the use of four times the number of contrastive blocks compared to the original TS2vec.

The results are presented in Table 9. We observe that simply increasing the number of epochs or contrastive blocks does not improve performance but rather leads to a decrease in most cases. We speculate that this decrease is caused by overfitting.

## Appendix H  Broader Impacts

Our approach for self-supervised contrastive learning improves classification performance on target datasets in patient-independent medical diagnosis scenarios. Leveraging different data consistency levels in medical time series is crucial to enable effective and accurate contrastive learning without sufficient labels. Our work will encourage the research community to discover universal frameworks for other practical applications based on time series representation learning. We also hope our work can attract more researchers to the more general problem of hierarchical consistency from other related fields.

From the societal perspective, our work and the line of contrastive learning can promote more efficient use of medical time series with the lack of labels. Specifically, our model has the potential to identify patterns and anomalies that may not be immediately apparent to human experts. This could lead to earlier and more accurate diagnoses, improving patient outcomes and reducing healthcare costs. However, practitioners need to be aware of the limitations of the model.

**Table 9: Heavy Duty Baselines**. Run more epochs or add more contrastive blocks to SimCLR and TS2vec on the AD dataset.

| Fraction | Models | Accuracy | Precision | Recall | F1 score | AUROC | AUPRC |
|---|---|---|---|---|---|---|---|
| **100%** | **4E-SimCLR** | $56.87_{\pm2.51}$ | $57.67_{\pm4.02}$ | $53.31_{\pm2.30}$ | $48.28_{\pm4.63}$ | $57.67_{\pm4.02}$ | $51.97_{\pm1.44}$ |
| | **4B-SimCLR** | $53.92_{\pm3.81}$ | $51.88_{\pm3.25}$ | $51.26_{\pm3.05}$ | $46.92_{\pm5.07}$ | $51.88_{\pm3.25}$ | $50.75_{\pm1.69}$ |
| | **SimCLR** | $54.77_{\pm1.97}$ | $50.15_{\pm7.02}$ | $50.58_{\pm1.92}$ | $43.18_{\pm4.27}$ | $50.15_{\pm7.02}$ | $50.42_{\pm1.06}$ |
| | **2E-TS2vec** | $76.49_{\pm6.10}$ | $78.97_{\pm3.54}$ | $77.69_{\pm5.16}$ | $76.21_{\pm6.45}$ | $88.44_{\pm2.40}$ | $88.12_{\pm2.53}$ |
| | **2B-TS2vec** | $81.61_{\pm1.65}$ | $81.47_{\pm1.75}$ | $81.53_{\pm1.68}$ | $81.43_{\pm1.65}$ | $89.50_{\pm1.60}$ | $89.22_{\pm1.75}$ |
| | **TS2vec** | $81.26_{\pm2.08}$ | $81.21_{\pm2.14}$ | $81.34_{\pm2.04}$ | $81.12_{\pm2.06}$ | $89.20_{\pm1.76}$ | $88.94_{\pm1.85}$ |
| | **COMET (Ours)** | $\mathbf{84.50_{\pm4.46}}$ | $\mathbf{88.31_{\pm2.42}}$ | $\mathbf{82.95_{\pm5.39}}$ | $\mathbf{83.33_{\pm5.15}}$ | $\mathbf{94.44_{\pm2.37}}$ | $\mathbf{94.43_{\pm2.48}}$ |
| **10%** | **4E-SimCLR** | $57.97_{\pm1.74}$ | $58.41_{\pm8.31}$ | $53.69_{\pm2.25}$ | $46.47_{\pm5.71}$ | $58.41_{\pm8.31}$ | $52.33_{\pm1.38}$ |
| | **4B-SimCLR** | $53.57_{\pm6.29}$ | $53.89_{\pm5.05}$ | $52.97_{\pm4.22}$ | $50.60_{\pm5.46}$ | $53.89_{\pm5.05}$ | $51.80_{\pm2.34}$ |
| | **SimCLR** | $56.09_{\pm2.25}$ | $53.81_{\pm5.74}$ | $51.73_{\pm2.59}$ | $44.10_{\pm4.84}$ | $53.81_{\pm5.74}$ | $51.08_{\pm1.53}$ |
| | **2E-TS2vec** | $66.29_{\pm7.86}$ | $69.92_{\pm5.13}$ | $67.79_{\pm6.44}$ | $65.36_{\pm8.55}$ | $78.54_{\pm4.98}$ | $77.95_{\pm5.29}$ |
| | **2B-TS2vec** | $72.61_{\pm4.46}$ | $73.86_{\pm4.36}$ | $72.98_{\pm3.66}$ | $72.30_{\pm4.24}$ | $81.91_{\pm4.83}$ | $81.74_{\pm4.85}$ |
| | **TS2vec** | $73.28_{\pm4.34}$ | $74.14_{\pm4.33}$ | $73.52_{\pm3.77}$ | $73.00_{\pm4.18}$ | $81.66_{\pm5.20}$ | $81.58_{\pm5.11}$ |
| | **COMET (Ours)** | $\mathbf{91.43_{\pm3.12}}$ | $\mathbf{92.52_{\pm2.36}}$ | $\mathbf{90.71_{\pm3.56}}$ | $\mathbf{91.14_{\pm3.31}}$ | $\mathbf{96.44_{\pm2.84}}$ | $\mathbf{96.48_{\pm2.82}}$ |
| **1%** | **4E-SimCLR** | $58.07_{\pm1.93}$ | $57.72_{\pm3.50}$ | $54.92_{\pm1.99}$ | $51.93_{\pm3.11}$ | $57.72_{\pm3.50}$ | $52.91_{\pm1.28}$ |
| | **4B-SimCLR** | $54.67_{\pm5.43}$ | $54.86_{\pm4.94}$ | $54.48_{\pm4.64}$ | $53.68_{\pm4.89}$ | $54.86_{\pm4.94}$ | $52.67_{\pm2.68}$ |
| | **SimCLR** | $55.42_{\pm2.43}$ | $52.18_{\pm5.55}$ | $51.37_{\pm2.76}$ | $45.02_{\pm4.79}$ | $52.18_{\pm5.55}$ | $50.87_{\pm1.45}$ |
| | **2E-TS2vec** | $63.56_{\pm4.62}$ | $64.97_{\pm3.53}$ | $64.49_{\pm3.90}$ | $63.28_{\pm4.69}$ | $70.26_{\pm3.55}$ | $68.77_{\pm3.59}$ |
| | **2B-TS2vec** | $64.18_{\pm4.53}$ | $64.26_{\pm4.80}$ | $64.26_{\pm4.80}$ | $63.93_{\pm4.61}$ | $70.07_{\pm5.97}$ | $68.62_{\pm6.25}$ |
| | **TS2vec** | $64.93_{\pm3.53}$ | $65.28_{\pm3.52}$ | $65.14_{\pm3.59}$ | $64.64_{\pm3.58}$ | $70.56_{\pm5.38}$ | $68.97_{\pm5.75}$ |
| | **COMET (Ours)** | $\mathbf{88.22_{\pm3.36}}$ | $\mathbf{88.55_{\pm2.73}}$ | $\mathbf{88.56_{\pm3.14}}$ | $\mathbf{88.14_{\pm3.37}}$ | $\mathbf{96.05_{\pm1.36}}$ | $\mathbf{96.12_{\pm1.31}}$ |

All datasets in this paper are publicly available and are not associated with privacy or security concerns. Furthermore, we have followed guidelines on responsible use specified by the primary authors of the datasets used in the current work.

# Appendix References

[S1] Cosimo Ieracitano, Nadia Mammone, Alessia Bramanti, Amir Hussain, and Francesco C Morabito. A convolutional neural network approach for classification of dementia stages based on 2d-spectral representation of eeg recordings. Neurocomputing, 323:96–107, 2019.

[S2] Ling Yang and Shenda Hong. Unsupervised time-series representation learning with iterative bilinear temporal-spectral fusion. In International Conference on Machine Learning, pages 25038–25054. PMLR, 2022.

[S3] Dennis Bäßler, Tobias Kortus, and Gabriele Gühring. Unsupervised anomaly detection in multivariate time series with online evolving spiking neural networks. Machine Learning, 111(4):1377–1408, 2022.

