| $81.69_{\pm10.71}$ | $87.28_{\pm5.71}$ | $79.64_{\pm12.07}$ | $78.90_{\pm13.78}$ | $92.54_{\pm4.18}$ | $92.36_{\pm4.34}$ |
| | | S | $83.53_{\pm2.89}$ | $84.35_{\pm1.69}$ | $82.81_{\pm3.56}$ | $82.97_{\pm3.47}$ | $91.01_{\pm1.96}$ | $90.81_{\pm1.95}$ |
| | | R | $78.58_{\pm14.99}$ | $83.14_{\pm11.60}$ | $76.31_{\pm16.66}$ | $74.53_{\pm18.69}$ | $85.05_{\pm12.74}$ | $84.44_{\pm13.22}$ |
| | | P | $72.69_{\pm13.13}$ | $78.67_{\pm11.81}$ | $69.74_{\pm14.50}$ | $67.04_{\pm17.27}$ | $83.30_{\pm13.91}$ | $82.57_{\pm14.29}$ |
| | | S+O | $\mathbf{85.70_{\pm1.82}}$ | $86.14_{\pm1.63}$ | $\mathbf{85.09_{\pm2.04}}$ | $\mathbf{85.35_{\pm1.96}}$ | $92.21_{\pm1.40}$ | $92.09_{\pm1.39}$ |
| | | R+S+O | $85.57_{\pm4.04}$ | $88.12_{\pm2.58}$ | $84.31_{\pm4.79}$ | $84.73_{\pm4.59}$ | $93.28_{\pm2.52}$ | $92.98_{\pm2.83}$ |
| | | P+R+S+O | $84.50_{\pm4.46}$ | $\mathbf{88.31_{\pm2.42}}$ | $82.95_{\pm5.39}$ | $83.33_{\pm5.15}$ | $\mathbf{94.44_{\pm2.37}}$ | $\mathbf{94.43_{\pm2.48}}$ |
| | 10% | O | $86.35_{\pm9.25}$ | $87.09_{\pm9.04}$ | $85.55_{\pm9.92}$ | $85.70_{\pm10.10}$ | $92.62_{\pm8.99}$ | $92.77_{\pm8.72}$ |
| | | S | $77.30_{\pm3.63}$ | $78.45_{\pm3.75}$ | $76.26_{\pm3.91}$ | $76.36_{\pm3.92}$ | $85.43_{\pm3.61}$ | $84.63_{\pm3.76}$ |
| | | R | $77.83_{\pm17.59}$ | $83.45_{\pm14.35}$ | $75.41_{\pm19.62}$ | $72.08_{\pm23.45}$ | $83.51_{\pm16.90}$ | $83.27_{\pm17.00}$ |
| | | P | $71.41_{\pm15.31}$ | $76.91_{\pm15.56}$ | $68.70_{\pm16.46}$ | $66.54_{\pm17.74}$ | $76.11_{\pm16.98}$ | $76.06_{\pm16.19}$ |
| | | S+O | $82.73_{\pm2.05}$ | $84.33_{\pm2.71}$ | $81.51_{\pm2.07}$ | $81.98_{\pm2.11}$ | $90.02_{\pm2.47}$ | $90.02_{\pm2.40}$ |
| | | R+S+O | $91.19_{\pm3.14}$ | $91.74_{\pm3.01}$ | $90.70_{\pm3.34}$ | $90.98_{\pm3.25}$ | $95.86_{\pm2.63}$ | $95.86_{\pm2.67}$ |
| | | P+R+S+O | $\mathbf{91.43_{\pm3.12}}$ | $\mathbf{92.52_{\pm2.36}}$ | $\mathbf{90.71_{\pm3.56}}$ | $\mathbf{91.14_{\pm3.31}}$ | $\mathbf{96.44_{\pm2.84}}$ | $\mathbf{96.48_{\pm2.82}}$ |
| | 1% | O | $69.56_{\pm7.54}$ | $71.43_{\pm8.04}$ | $68.19_{\pm7.16}$ | $67.83_{\pm7.41}$ | $77.32_{\pm8.69}$ | $77.32_{\pm8.45}$ |
| | | S | $59.09_{\pm3.50}$ | $61.08_{\pm4.86}$ | $59.83_{\pm4.00}$ | $58.12_{\pm3.57}$ | $63.99_{\pm6.62}$ | $63.33_{\pm6.09}$ |
| | | R | $\mathbf{90.15_{\pm13.23}}$ | $\mathbf{93.83_{\pm7.28}}$ | $\mathbf{89.03_{\pm14.88}}$ | $\mathbf{88.17_{\pm16.78}}$ | $\mathbf{96.22_{\pm5.97}}$ | $96.02_{\pm6.28}$ |
| | | P | $85.57_{\pm13.45}$ | $90.36_{\pm7.16}$ | $83.98_{\pm15.13}$ | $82.72_{\pm18.03}$ | $94.23_{\pm5.82}$ | $94.10_{\pm5.65}$ |
| | | S+O | $63.24_{\pm3.62}$ | $63.52_{\pm4.15}$ | $63.30_{\pm4.26}$ | $62.72_{\pm4.14}$ | $67.98_{\pm5.17}$ | $67.25_{\pm4.97}$ |
| | | R+S+O | $82.65_{\pm4.23}$ | $83.21_{\pm4.06}$ | $82.97_{\pm4.58}$ | $82.46_{\pm4.43}$ | $90.07_{\pm6.77}$ | $90.24_{\pm6.56}$ |
| | | P+R+S+O | $88.22_{\pm3.36}$ | $88.55_{\pm2.73}$ | $88.56_{\pm3.14}$ | $88.14_{\pm3.37}$ | $96.05_{\pm1.36}$ | $\mathbf{96.12_{\pm1.31}}$ |
| PTB | 100% | O | $85.27_{\pm1.73}$ | $84.21_{\pm1.25}$ | $77.74_{\pm4.01}$ | $79.78_{\pm3.37}$ | $88.13_{\pm2.23}$ | $84.76_{\pm2.14}$ |
| | | S | $84.30_{\pm2.56}$ | $84.00_{\pm2.16}$ | $75.68_{\pm5.69}$ | $77.74_{\pm5.13}$ | $88.66_{\pm2.05}$ | $84.80_{\pm2.15}$ |
| | | R | $88.63_{\pm1.43}$ | $88.42_{\pm1.45}$ | $82.46_{\pm2.35}$ | $84.72_{\pm2.12}$ | $89.29_{\pm3.96}$ | $86.21_{\pm4.96}$ |
| | | P | $\mathbf{88.85_{\pm3.22}}$ | $\mathbf{88.74_{\pm2.30}}$ | $82.78_{\pm6.11}$ | $84.75_{\pm5.38}$ | $\mathbf{94.32_{\pm1.81}}$ | $\mathbf{90.61_{\pm2.81}}$ |
| | | S+O | $84.38_{\pm2.34}$ | $84.33_{\pm1.18}$ | $75.49_{\pm5.69}$ | $77.69_{\pm4.76}$ | $90.37_{\pm2.59}$ | $86.57_{\pm4.32}$ |
| | | R+S+O | $85.32_{\pm1.93}$ | $84.82_{\pm1.78}$ | $77.54_{\pm4.58}$ | $79.64_{\pm3.94}$ | $92.34_{\pm1.85}$ | $88.75_{\pm1.98}$ |
| | | P+R+S+O | $86.36_{\pm1.44}$ | $87.18_{\pm1.28}$ | $77.87_{\pm2.55}$ | $80.79_{\pm2.44}$ | $93.41_{\pm2.35}$ | $89.09_{\pm1.64}$ |
| | 10% | O | $86.75_{\pm1.44}$ | $85.42_{\pm2.10}$ | $80.88_{\pm3.27}$ | $82.45_{\pm2.29}$ | $90.71_{\pm3.16}$ | $88.84_{\pm3.59}$ |
| | | S | $86.34_{\pm0.73}$ | $84.75_{\pm1.49}$ | $80.21_{\pm1.78}$ | $81.90_{\pm1.25}$ | $88.36_{\pm0.31}$ | $85.20_{\pm1.17}$ |
| | | R | $88.12_{\pm1.75}$ | $86.36_{\pm2.28}$ | $84.16_{\pm3.55}$ | $84.80_{\pm2.39}$ | $91.80_{\pm2.50}$ | $90.45_{\pm1.90}$ |
| | | P | $\mathbf{90.38_{\pm2.23}}$ | $\mathbf{90.33_{\pm2.71}}$ | $85.69_{\pm4.36}$ | $87.27_{\pm3.45}$ | $\mathbf{93.06_{\pm3.86}}$ | $\mathbf{91.83_{\pm2.93}}$ |
| | | S+O | $85.84_{\pm1.74}$ | $84.14_{\pm0.76}$ | $80.07_{\pm5.18}$ | $81.14_{\pm3.82}$ | $90.28_{\pm2.75}$ | $87.57_{\pm3.42}$ |
| | | R+S+O | $85.88_{\pm3.08}$ | $83.60_{\pm3.61}$ | $82.45_{\pm1.94}$ | $82.48_{\pm2.71}$ | $90.79_{\pm1.79}$ | $88.68_{\pm1.91}$ |
| | | P+R+S+O | $90.32_{\pm1.61}$ | $89.67_{\pm1.94}$ | $\mathbf{85.85_{\pm2.99}}$ | $\mathbf{87.35_{\pm2.36}}$ | $92.78_{\pm1.11}$ | $90.46_{\pm3.09}$ |
| | 1% | O | $85.73_{\pm1.59}$ | $83.79_{\pm2.15}$ | $79.53_{\pm3.14}$ | $81.09_{\pm2.54}$ | $90.15_{\pm2.32}$ | $88.85_{\pm2.64}$ |
| | | S | $77.63_{\pm1.72}$ | $72.36_{\pm2.41}$ | $69.41_{\pm2.53}$ | $70.38_{\pm2.29}$ | $79.09_{\pm2.21}$ | $75.07_{\pm1.88}$ |
| | | R | $86.28_{\pm1.97}$ | $84.02_{\pm2.85}$ | $82.06_{\pm2.08}$ | $82.61_{\pm1.97}$ | $90.50_{\pm0.99}$ | $88.32_{\pm1.40}$ |
| | | P | $\mathbf{91.55_{\pm2.45}}$ | $\mathbf{90.28_{\pm3.27}}$ | $\mathbf{88.45_{\pm3.22}}$ | $\mathbf{89.25_{\pm3.13}}$ | $\mathbf{95.12_{\pm2.58}}$ | $\mathbf{94.80_{\pm2.04}}$ |
| | | S+O | $81.78_{\pm1.07}$ | $77.65_{\pm1.34}$ | $77.17_{\pm2.14}$ | $77.20_{\pm1.43}$ | $86.65_{\pm2.38}$ | $82.12_{\pm1.95}$ |
| | | R+S+O | $80.71_{\pm2.28}$ | $76.79_{\pm2.03}$ | $80.51_{\pm1.67}$ | $77.84_{\pm2.13}$ | $88.76_{\pm1.38}$ | $85.00_{\pm2.23}$ |
| | | P+R+S+O | $86.62_{\pm3.90}$ | $82.99_{\pm4.52}$ | $85.36_{\pm5.73}$ | $83.89_{\pm4.96}$ | $93.78_{\pm3.82}$ | $91.94_{\pm4.96}$ |
| TDBrain | 100% | O | $89.92_{\pm2.15}$ | $90.50_{\pm2.07}$ | $89.92_{\pm2.15}$ | $89.89_{\pm2.17}$ | $96.79_{\pm1.29}$ | $96.84_{\pm1.29}$ |
| | | S | $77.86_{\pm2.85}$ | $78.98_{\pm2.51}$ | $77.86_{\pm2.85}$ | $77.62_{\pm3.01}$ | $88.44_{\pm2.23}$ | $88.96_{\pm2.14}$ |
| | | R | $\mathbf{94.44_{\pm1.79}}$ | $\mathbf{94.60_{\pm1.75}}$ | $\mathbf{94.44_{\pm1.79}}$ | $\mathbf{94.44_{\pm1.80}}$ | $\mathbf{98.39_{\pm1.29}}$ | $\mathbf{98.31_{\pm1.46}}$ |
| | | P | $93.18_{\pm3.70}$ | $93.27_{\pm3.67}$ | $93.18_{\pm3.70}$ | $93.18_{\pm3.70}$ | $97.59_{\pm2.01}$ | $97.58_{\pm2.00}$ |
| | | S+O | $80.38_{\pm4.22}$ | $81.59_{\pm3.70}$ | $80.38_{\pm4.22}$ | $80.16_{\pm4.38}$ | $91.35_{\pm2.24}$ | $91.64_{\pm2.05}$ |
| | | R+S+O | $86.01_{\pm4.29}$ | $86.35_{\pm3.81}$ | $86.01_{\pm4.29}$ | $85.95_{\pm4.38}$ | $94.14_{\pm2.72}$ | $94.37_{\pm2.58}$ |
| | | P+R+S+O | $85.47_{\pm1.16}$ | $85.68_{\pm1.20}$ | $85.47_{\pm1.16}$ | $85.45_{\pm1.16}$ | $93.73_{\pm1.02}$ | $93.96_{\pm0.99}$ |
| | 10% | O | $85.34_{\pm4.38}$ | $86.03_{\pm3.97}$ | $85.34_{\pm4.38}$ | $85.25_{\pm4.48}$ | $93.18_{\pm3.52}$ | $93.24_{\pm3.53}$ |
| | | S | $74.02_{\pm2.09}$ | $74.62_{\pm2.01}$ | $74.02_{\pm2.09}$ | $73.86_{\pm2.17}$ | $81.92_{\pm3.25}$ | $81.76_{\pm3.42}$ |
| | | R | $\mathbf{92.96_{\pm8.22}}$ | $\mathbf{93.02_{\pm8.20}}$ | $\mathbf{92.96_{\pm8.22}}$ | $\mathbf{92.96_{\pm8.23}}$ | $\mathbf{96.14_{\pm5.80}}$ | $\mathbf{95.93_{\pm6.08}}$ |
| | | P | $89.38_{\pm14.69}$ | $89.58_{\pm14.72}$ | $89.38_{\pm14.69}$ | $89.36_{\pm14.69}$ | $93.23_{\pm11.81}$ | $93.52_{\pm11.10}$ |
| | | S+O | $74.92_{\pm2.57}$ | $76.60_{\pm3.07}$ | $74.92_{\pm2.57}$ | $74.54_{\pm2.55}$ | $84.51_{\pm3.07}$ | $84.40_{\pm3.07}$ |
| | | R+S+O | $81.90_{\pm4.74}$ | $83.55_{\pm4.02}$ | $81.90_{\pm4.74}$ | $81.61_{\pm4.99}$ | $91.21_{\pm3.52}$ | $90.73_{\pm3.72}$ |
| | | P+R+S+O | $79.28_{\pm4.64}$ | $79.83_{\pm4.83}$ | $79.28_{\pm4.64}$ | $79.19_{\pm4.62}$ | $88.39_{\pm4.13}$ | $88.38_{\pm3.96}$ |
| | 1% | O | $71.52_{\pm7.54}$ | $72.17_{\pm7.71}$ | $71.52_{\pm7.54}$ | $71.31_{\pm7.62}$ | $78.32_{\pm8.37}$ | $77.56_{\pm8.91}$ |
| | | S | $58.62_{\pm3.84}$ | $59.66_{\pm3.95}$ | $58.62_{\pm3.84}$ | $57.39_{\pm4.26}$ | $62.85_{\pm6.38}$ | $61.61_{\pm6.61}$ |
| | | R | $\mathbf{85.29_{\pm5.93}}$ | $\mathbf{85.55_{\pm5.66}}$ | $\mathbf{85.29_{\pm5.93}}$ | $\mathbf{85.24_{\pm5.99}}$ | $\mathbf{91.19_{\pm4.76}}$ | $\mathbf{91.11_{\pm4.74}}$ |
| | | P | $77.23_{\pm3.42}$ | $78.10_{\pm3.39}$ | $77.23_{\pm3.42}$ | $77.04_{\pm3.47}$ | $86.12_{\pm3.88}$ | $85.10_{\pm4.11}$ |
| | | S+O | $61.71_{\pm2.97}$ | $61.82_{\pm2.90}$ | $61.71_{\pm2.97}$ | $61.60_{\pm3.07}$ | $66.09_{\pm2.98}$ | $64.94_{\pm2.66}$ |
| | | R+S+O | $72.15_{\pm6.26}$ | $73.39_{\pm7.50}$ | $72.15_{\pm6.26}$ | $71.91_{\pm6.12}$ | $78.39_{\pm7.77}$ | $76.97_{\pm7.61}$ |
| | | P+R+S+O | $72.93_{\pm7.21}$ | $74.20_{\pm7.68}$ | $72.93_{\pm7.21}$ | $72.57_{\pm7.37}$ | $78.72_{\pm8.42}$ | $77.71_{\pm9.10}$ |