# OpenReview forum: "Contrast Everything: A Hierarchical Contrastive Framework for Medical Time-Series"
_NeurIPS.cc/2023/Conference — NeurIPS 2023 poster_

### Official Review · Reviewer_p5Bk · 2023-06-30

**Soundness:** 3 good
**Presentation:** 3 good
**Contribution:** 3 good
**Rating:** 6
**Confidence:** 4

**Summary:**

This paper investigates the contrastive learning for medical time series and develops multiple contrastive objectives at different levels (patient, trial, observation, sample). The downstream applications is the classification on EEG, ECG, EMG, EOG signals. The proposed method has obtained the best or competitive performance on two EEG datasets and one ECG dataset.

**Strengths:**

- It is interesting to see a hierarchical view for a specific type of medical time series, such as ECG and EEG. This hierarchical multi-level view captures the unique characteristics of this type of time series.
- Given this novel view, the proposed contrastive learning at all levels is straightforward and reasonable.
- The paper is well written and easy to understand.


**Weaknesses:**

- My first concern is the discrepancy between the title and the actual methodology. "multi-level contrastive learning" sounds a better choice than "multi-granularity representation learning" because event at the patient or trial level, the representation learning still happens at the sample-level input, and you do not squeeze or aggregate representations at higher levels.Thus, I feel there is no multiple granularity. Please correct me if I misunderstand your method.
- The second concern is that the proposed method is only evaluated on limited datasets. It seems that the method only works on EEG-related datasets (AD and TDBrain), beating other baselines significantly, but is modest on PTB, an ECG dataset. I would like to see broader empirical demonstrations and analyses.
- Moreover, the ablation tests are only conducted on the AD dataset, on which the proposed method has achieved the most performance improvements. I would like to see more detailed ablation tests and analyses on other datasets.

**Questions:**

Check the weakness part.

**Limitations:**

Lack of sufficient verifications on large-scale and more diversified datatsets.

---

> ### Author Rebuttal · Authors · 2023-08-09
>
> We are happy that you are interested in our paper and feel our paper is well-written and easy to understand. Thank you very much for your three valuable concerns! Here we respond in detail to each of your concerns. If you do not feel we have sufficiently justified a higher score, please let us know where we can improve our work and your further concern. Thank you again!
> ***
> **Q1**: My first concern is the discrepancy between the title and the actual methodology. "multi-level contrastive learning" sounds like a better choice than "multi-granularity representation learning" because even at the patient or trial level, the representation learning still happens at the sample-level input, and you do not squeeze or aggregate representations at higher levels. Thus, I feel there is no multiple granularity. Please correct me if I misunderstand your method.
>
> **A1**: Thanks for bringing up this concern. We acknowledge that we do not compress or aggregate representations at higher levels; the representation learning consistently occurs at the sample-level input.
>
> However, our objective is not to acquire patient or trial level representations. Instead, the various granularities serve to construct positive and negative pairs for contrastive learning. By harnessing patient and trial level information, we extend our representation learning beyond conventional instance discrimination methods. This approach facilitates capturing consistencies across samples (instances) through a self-supervised framework. In the original submission, we have taken cognizance of your concern and examined the difference between our method and multi-granularity approaches in other domains. Please refer to Appendix E.2 for a detailed analysis.
>
> Moreover, we appreciate your attention to the paper's title. We will thoroughly consider whether to modify it in the camera-ready version. Indeed, we deliberated on the paper's title extensively. Alternately, we contemplated a title such as "Hierarchical Contrastive Learning." What are your thoughts on this title?
> ***
> **Q2**: The second concern is that the proposed method is only evaluated on limited datasets. It seems that the method only works on EEG-related datasets (AD and TDBrain), beating other baselines significantly, but is modest on PTB, an ECG dataset. I would like to see broader empirical demonstrations and analyses.
>
> **A2**: Thank you for raising this concern. We add one more large-scale ECG dataset (PTB-XL; 17596 patients, 191400 samples, 5 classes). See R-table 3 for experiment results. Our method performs better in 11 of 12 tests than the other three SOTAs. Particularly with label fraction 1%, we outperform SOTAs by about 5% F1 score and 3% AUROC, demonstrating the effectiveness of contrastive pre-train of COMET in reducing the dependence of labeled data.
>
> ***
> **Q3**: Moreover, the ablation tests are only conducted on the AD dataset, on which the proposed method has achieved the most performance improvements. I would like to see more detailed ablation tests and analyses on other datasets.
>
> **A3**: Thank you for raising this concern. We add the ablation study on the TDBrain and PTB. See R-table 1 and 2 for experiment results.

---

> > ### Comment · Reviewer_p5Bk · 2023-08-17
> > **Thanks for the reply**
> >
> > "Hierarchical contrastive learning" better aligns with the underlying approach of this paper. This paper has nothing to do with multi-granularity representations. Using "multi-granularity representation learning" can be very misleading.
> >
> > More datasets and more ablation results look good.
> >
> > I would only like to raise the score to "weakly accept" because the current version needs to be modified a lot to fit for the new theme, "hierarchical contrastive learning".

---

> > > ### Author Response · Authors · 2023-08-17
> > > **Thanks for the reply**
> > >
> > > Thank you again for improving the score! Thank you for your constructive suggestion on our paper’s title. We will change our title to “Hierarchical Contrastive Learning for Medical Time-Series.” Besides,  we will thoroughly modify our terms/notations/figures/text to keep the paper consistent and easy to follow. For example,  we plan to update all “granularity” into “level” to reduce the potential misunderstanding caused by “granularity” in the camera-ready version.

---

### Official Review · Reviewer_jbn1 · 2023-07-02

**Soundness:** 3 good
**Presentation:** 3 good
**Contribution:** 2 fair
**Rating:** 5
**Confidence:** 4

**Summary:**

This paper proposes a multi-granularity contrastive learning method (named COMET) on medical time-series data. The methods build postive and negative pairs from patient-level, trail-level, sample-level, and observation-level. The proposed method is evaluated on two EEG, and one ECG dataset.

**Strengths:**

1. The paper is well organized and clearly written. Figure 2 is very informative. Related works are also comprehensive.
2. The performance of the proposed method is decent based on the experimental tables.

**Weaknesses:**

1. The proposed method is not new and many multi-granular contrastive learning models have been proposed in the past few years. Based on Figure 1, CLOCS, TS2Vec, TS-TCC are also multi-granular contrastive learning methods specifically in medical timeseries domain. The contribution and extension (from existing two granular methods to the proposed four granular method) of this paper seems trivial given these exisiting methods.
2. Overall, the proposed COMET method can be viewed as "a more finer-granular way of doing data augmentation". In that sense, the data augmentation methods for all models (including baselines) seem different, then does the experimental design fair?
3. The considered two EEG and one ECG datasets all seems small in medical timeseries domain. What is the number of parameters in the model? Usually, the training data size is expected to be 10 times larger than the number of parameters. Contrastive learning methods naturally need a lot more unlabeled data to train. SHHS is a large EEG dataset, and the ECG 2020 challenge datasets are also large https://physionet.org/content/challenge-2020/1.0.2/.
4. Although the authors have gave another example on satellite sensor applications, the generalizability of the proposed model is still questionable. Given that many multi-granular contrastive learning models have been developed in other domains as well, could the authors distinguish this paper and highlights the key features of COMET if generalizable to other domains?

**Questions:**

see above

---

> ### Author Rebuttal · Authors · 2023-08-09
>
> We are grateful for your thoughtful feedback and happy that you feel our paper is well-organized and the figure is informative. The following response to the questions about our method's weaknesses.
> ***
> **Q1**: The proposed method is not new since many multi-granular contrastive learning models exist.
>
> **A1**: We agree that others have implicitly utilized multi-level information, albeit within highly specific models tailored to different data types. In contrast, we introduce a general framework that applies to all types of medical time series data. We explicitly present the concept of multi-granularity in the context of contrastive learning.
>
> After reviewing numerous existing contrastive learning methods within the time series domain, we consistently posed a pivotal question to ourselves: Can we design a straightforward yet appliable contrastive learning framework that can be adapted to all forms of medical time series data, akin to the classical model SimCLR in the domain of contrastive learning? In contrast to general time series data, medical time series data typically exhibit two additional granularities: patient and trial. Our objective is to craft an innovative framework that utilizes all information within medical time series in a self-supervised manner. This approach enables us to harness the information of patient and trial level to learn consistency across instances. Simultaneously, we leverage the sample and observation levels to facilitate conventional instance discrimination. In COMET, users can easily toggle between different granularities or adjust their weights by setting the hyper-parameter λ, depending on the distinctive features of the medical time series.
>
> Some existing methods like TS2Vec, Clocs, TS-TCC are also multi-granularity. They can be viewed as a particular case of our model. In other words, we are not an extension of their model, but rather, in the domain of medical time series, our approach includes and is compatible with their model. Our work will play a role in summarizing, inspiring, and guiding future works in contrastive learning on medical time series.
>
> ***
> **Q2**: If the data augmentation methods for all models (including baselines) seem different, then does the experimental design fair?
>
> **A2**: As mentioned in A1, our model is not an extension but rather includes and is compatible with their models within the domain of medical time series. Additionally, existing methods like TFC, TS2Vec, and TimeCLR [10] have their specific data augmentation techniques integrated as part of their methods. Therefore, their original data augmentation methods should remain unchanged when using them as baselines for comparison.
>
> In our original submission, considering the extra contrastive blocks in our method, we did the heavy-duty case study by running extra epochs during contrastive pre-training or implementing extra contrastive blocks for baselines. (Appendix F.4)
>
>
> ***
>
> **Q3**: The two EEG and one ECG datasets seem small in the medical time series domain. The training data size is expected to be 10 times larger than the number of parameters. Use SHHS or the ECG 2020 challenge datasets.
>
> **A3**: Our responses to this question contain three parts.
>
> 1) Regarding the PTB dataset, our model comprises 687,552 parameters. While increasing the size of the training data to exceed the number of parameters is indeed an effective approach to counter overfitting, modern techniques such as L2 regularization, dropout, and batch normalization have proven useful in mitigating overfitting risks, enabling models to yield favorable performance even with limited data [11][12].
>
> 2) Our datasets are generally not small within the scope of medical time series. The medical time series domain faces challenges in labeling due to factors like the complexity of data collection and associated expenses, including issues such as ethnic considerations [1][2]. Moreover, datasets centered around specific ailments, such as Alzheimer's disease, are rarely accessible and often not publicly available [7].
>
> 3) In the original submission, we utilized the PTB dataset, part of the ECG 2020 challenge available at Physionet 2020. Additionally, we newly add a considerably larger dataset  PTB-XL (17596 patients, 191400 samples, 5 classes) within the ECG 2020 challenge. See R-table 3. Our method performs better in 11 of 12 tests than the other three SOTAs. Particularly with label fraction 1%, we outperform SOTAs by about 5% F1 score and 3% AUROC, demonstrating the effectiveness of contrastive pre-train of COMET in reducing the dependence of labeled data.
>
> ***
> **Q4**: The generalizability of the proposed model?
>
> **A4**: Our response has three parts.
>
> 1) As indicated in the title and abstract, COMET is uniquely tailored for medical time series, capitalizing on their distinctive characteristics. It is not a universally applicable model for all types of time series, as general time series typically only have sample and observation granularities.
>
> 2) We offer an illustrative example involving satellite sensor applications, intending to inspire researchers in diverse domains. The key is to utilize all available information, excluding label data, for contrastive pre-training, such as patient ID. To adapt our approach to other domains, researchers must consider a crucial question: Does the dataset has additional information beyond sample labels? If affirmative, can this information be harnessed for contrastive learning? The example of satellite sensor application underscores the potential existence of supplementary information even in non-medical domains.
>
> 3) There are also other multi-granularity papers in different domains. We review and compare them with our methods in Appendix E.2. Some papers focus on learning representations of various granularities, which differs from ours. In our case, the distinct granularities are employed for constructing positive and negative pairs during contrastive learning.

---

### Official Review · Reviewer_RVrY · 2023-07-03

**Soundness:** 3 good
**Presentation:** 4 excellent
**Contribution:** 3 good
**Rating:** 8
**Confidence:** 5

**Summary:**

Medical time-series data, unlike domains such as CV and NLP, lack data labeling but contained more layers of information corresponding to observation, trial, and individual physiologies. Unlike previous methods that overlooked the multi-granularity, the authors proposed COMET to trained the combined contrastive losses from all 4 granularities and presented COMET’s elevated performance on multiple datasets, outperforming many up-to-date contrastive baselines. With ablation studies and disclosed code repo, the authors evaluated the effectiveness of proposed COMET and concluded it as a pioneering framework tailored for medical time series.

**Strengths:**

•	Figures are well done and make it much easier to understand what the authors are talking about when they mention the different levels of granularity for medical time series data
•	The paper is very well written and easy to understand. The method seems simple but this is likely due to the authors’ explanation of information granularity. The paper follows a clear logical flow.
•	Capturing multi-granularity information with contrastive learning is a very good idea backed by performance improvements.
•	They backed their proposed COMET with comprehensive experiments on multiple datasets with many up-to-date baselines
•	The performance of their method is very high in comparison the methods they consider to be related work.


**Weaknesses:**

•	While the generalization is likely to follow from this work - this method is somewhat limited to ECG data. ECG data has many structural patterns in medical time series data that come from the cyclical nature of a cardiac cycle that other medical time series might not. As a result, the methods might actually be limited to pulsatile-type medical time series, which would still be a contribution.
For example On page 2, line 71, the statement that medical time series is "low cost and non-invasive." would not necessarily be correct for data collected from invasive A-line, for example. However, I mostly agree with the paragraph on EEG, ECG, etc., and how they benefit healthcare practices. I would suggest authors better approach this limitation which does not take away from the contribution of the work overall.
•	While many works are compared, other ECG-related methods, one of which they mention in the paper, CLOCs have not been compared. How does this method compare to ECG specific methods?
•	It was unclear how the time series features were fed into the COMET and how they were broken into different granularity and evaluated by different contrastive losses (see first bulletins of questions).
•	It would be better to include ablation study on the process of discovering the optimal set of hyper-coefficients for each dataset.
•	On Page 6 line 215, D is defined as dataset, but it should be better defined before referenced, namely in Section 4.2 or earlier?


**Questions:**

•	Can authors address the limitations? is this work specific to cyclical time-series data?
•	How do the results compare to other ECG contrastive pretraining methods?
•	Reading the “comet.py” and “models/encoder.py’s TSEncoder”, it seemed like the same inputs “x” were used to calculate 4 levels of contrastive losses. Then what exactly is “x” (the time series input of the COMET), is it observation-level? Sample-level? Can authors better describe the actual inputs to the system?



**Limitations:**

•	Only addresses one modality of medical timeseries data
•	Does not compare to other ECG contrastive pretraining methods

These have not been adequately addressed.

---

> ### Author Rebuttal · Authors · 2023-08-09
>
> Thank you for your very thoughtful feedback. We highly appreciate you carefully reading our figure, method, appendix, and code! We are happy you feel our paper follows a clear logic and is easy to follow!! Again, thank you for your comments to help us improve our paper. Here are the response to your questions.
> ***
>
> **Q1**: Can authors address the limitations? is this work specific to cyclical time-series data? • How do the results compare to other ECG contrastive pretraining methods?
>
> **A1**: Thank you for raising these questions. We add CLOCS and NCL as new baselines for comparison and add one more large-scale ECG dataset PTB-XL (17596 patients, 191400 samples, 5 classes)for the experiment. See R-table 3 for experiment results. Our method performs better in 11 of 12 tests than the other three SOTAs. Particularly with label fraction 1%, we outperform SOTAs by about 5% F1 score and 3% AUROC, demonstrating the effectiveness of contrastive pre-train of COMET in reducing the dependence of labeled data.
> ***
>
> **Q2**: Reading the “comet.py” and “models/encoder.py’s TSEncoder”, it seemed like the same inputs “x” were used to calculate 4 levels of contrastive losses. Then what exactly is “x” (the time series input of the COMET), is it observation-level? Sample-level? Can authors better describe the actual inputs to the system?
>
> **A2**: Thank you again for carefully reading our code! The input "x" utilized for each contrastive block is all sample-level data, as our primary objective entails acquiring sample-level representations.  A batch of sample-level data "x" is passed to the encoder to process and apply data augmentation to generate embeddings for each block. Distinct levels exhibit varying data augmentation approaches or no augmentation. Subsequently, the embeddings from each level, coupled with their respective patient/trial IDs, are fed into the system to compute the contrastive loss. Each level is strategically designed to capitalize on consistency at differing granularities, engendering distinct contrastive loss formulations. For instance, positive and negative pairs are generated through data augmentation at the observation and sample levels, whereas at the patient/trial level, they are formed based on the presence of the same patient/trial ID.

---

> > ### Comment · Reviewer_RVrY · 2023-08-15
> >
> > Thank you for your comments. I think these clarifications - and the new baselines - are valuable additions.

---

### Official Review · Reviewer_Trt3 · 2023-07-07

**Soundness:** 2 fair
**Presentation:** 3 good
**Contribution:** 2 fair
**Rating:** 4
**Confidence:** 5

**Summary:**

This paper proposes a multi-granularity framework leveraging data consistencies at different levels inherent in medical time series data. The model learns with contrastive loss designed at every data granularity, i.e., observation, sample, trial, and patient levels. The method is evaluated with three binary classification downstream tasks, showing significant gain over SOTA timeseries contrastive learning methods.

**Strengths:**

1. The idea of leveraging data consistencies are multi-granularity is interesting and seemingly effective.
2. The writing and presentation is easy to follow.


**Weaknesses:**

1. The contrastive learning for each granularity is simple and straight forward, making the novelty of the paper limited.
2. Majority of the medical tasks are fine-grained in nature. That is to say, the difference between two samples of difference classes may be very subtle and local. It is unclear how the data augmentation method for each granularity is designed to ensure not disturbing the sample label.
3. The evaluation is a bit unpersuasive. The downstream tasks are all simple binary classification problems (e.g. for dementia detection, Myocardial infarction, or Parkinson’s disease).
4. For medical diagnosis tasks, accuracy and F1 shall not be used as the main metric as the data distribution is usually drastically imbalance. I would suggest either focus on AUROC or AUPRC. In this case, COMET shows inferior performance on PTB than TS2vec. Any explanation for this results?


**Questions:**

1. The difference between ‘sample’ and ‘trial’ are a bit unclear. It seems that we can also consider a trail as a sample with a longer time span? How to segment trials into distinct samples? ”Each sample is a one-second interval with 256 observations. ” – Any justification for choosing one-second interval here?
2. I am curious about the implementation details for Observation-level data consistency. How $t^-$ is chosen? Shall it be close to $t$ or distant from $t$? How to guarantee that $x_{i, t^-}$ is more different to $x_{i, t}$ than $\tilde{x}_{i, t}$, as the time series data often display repetitive periodicity?
3. When jittering and masking are used for data augmentation, would it broken the continuity of the normal time series data?
4. The decision of using InfoNCE at observation and sample blocks while using NT-Xent loss at trial and patient level seems to be very arbitrary. Any explanation for it?
5. SOTA comparison: why NCL and CLOCS are not considered?
6. Any ambulation study on the hyperparameters $\lambda_i$ is missing.


**Limitations:**

the limitations are presented in the appendix.

---

> ### Author Rebuttal · Authors · 2023-08-09
>
> We are happy you feel our paper is easy to follow, interesting, and effective. We briefly respond to your concerns point-by-point. If you do not feel we have sufficiently justified a higher score, please let us know where we can further improve our work. Thank you again!
> ***
> **Q1** The contrastive learning for each granularity is simple and straightforward, limiting the novelty.
>
> **A1** We agree that contrastive learning of each granularity is straightforward and not overcomplicated. However, we sincerely believe that proposing a simple solution to address a challenging problem should exhibit the strengths of our work. In this paper, we posited the granularity conceptions in medical time series and designed a simple-yet-effective framework that uses all possible granularities in a self-supervised manner. Our COMET covers and is compatible with existing SOTAs on different granularities, which will play a significant role in summarizing, inspiring, and guiding future works in contrastive learning of medical time series.
>
> ***
> **Q2** The difference between two samples of different classes may be subtle and local. How does the data augmentation method for each granularity not disturb the sample label?
>
> **A2** Our answer contains two parts.
>
> 1) We did not apply data augmentation to patient and trial levels. Instead, we employed patient and trial IDs to establish positive and negative pairs for learning consistency across instances (Lines 167-176).
>
> 2) We used timestamp masking [6] for data augmentation at sample and observation levels. To prevent useless masking on zero-value regions, we applied masking to the projected embedding rather than the raw data (Lines 278-283). This method also mitigates the risk of masking affecting sample labels in medical tasks, as the embeddings of two samples in different classes could exhibit substantial differences after projection, even if their raw data appears similar.
>
> ***
> **Q3** The evaluation is a bit unpersuasive. The downstream tasks are all simple binary classification problems.
>
> **A3** Our answer contains three parts.
>
> 1) In the initial submission, we evaluated COMET on diverse downstream tasks besides binary classification, including clustering and anomaly detection (Appendix F.3).
>
> 2) We mainly presented the results of binary problems because the majority of real-world applications of medical time series for disease diagnosis are binary classification tasks [7][8].
>
> 3) We newly added a large-scale dataset PTB-XL (17596 patients, 191400 samples, 5 classes) to prove our model works well on multi-classification problems. See R-table 3. Our method performs better in 11 of 12 tests than the other three SOTAs. Particularly with label fraction 1%, we outperform SOTAs by about 5% F1 score and 3% AUROC, demonstrating the effectiveness of contrastive pre-train of COMET in reducing the dependence of labeled data.
> ***
> **Q4** 1) Suggest focusing on AUROC or AUPRC on imbalanced datasets. 2) COMET shows inferior performance on PTB than TS2vec.
>
> **A4**
> 1) We agree that AUROC and AUPRC are more appropriate for medical tasks. We presented six metrics (including AUROC and AUPRC) to provide comprehensive results to readers so they can analyze the results with their preferences. We will emphasize more about AUROC in the camera-ready version.
>
> 2) We did a careful ablation study on each block as you suggested and found increasing the lambda weight on the observation block benefits the performance (See R-table 2). We think the periodicity of ECG signals causes the reason. After adding more weight to the observation block, we showed better results on AUROC(93.67±2.34) compared with TS2Vec.
> ***
> **Q5** 1)The difference between ‘sample’ and ‘trial’ is unclear. Is the trial a sample with a longer time span? 2) How to segment trials into distinct samples? 3) Any justification for choosing a 1-second interval?
>
> **A5** Our answer contains three parts.
>
> 1) A trial can indeed be considered a sufficiently long sample. In the original submission, we have provided a detailed example in Appendix A to elucidate the distinctions between trial and sample. A trial is usually a much longer sample. It can span hours or even days, containing hundreds of thousands of timestamps, which is impractical for models to train on entire trials effectively. Consequently, researchers commonly partition them into shorter samples for training purposes [1][2][3].
>
> 2) Signal segmentation has been a standard operation in signal processing for decades. Our approach to segmenting each dataset is detailed in Appendix C.
>
> 3) The window size can be regarded as a hyper-parameter. Empirically, this selection ranges from 1 to 10 seconds, contingent on factors such as sampling frequency, data types, etc.[2][3]. We opt for 1 second simply due to its superior performance.
>
>
> ***
> **Q6** Implementation details for Observation-level data consistency. How $t^-$ is chosen? How to guarantee that $x_{i,t^-}$ is more different to $x_{i,t}$ than $\widetilde{x}_{i,t}$, as time series data often display repetitive periodicity?
>
> **A6** Our answer contains three parts.
>
> 1) Only observations augmented from the same raw observation are treated as positive pairs, while all observations with distinct timestamps within a sample are designated as $t^-$ (Fig. 2; Sec. 4.2). The principle to select negative observations ($x_{i,t^-}$) aligns with the majority of contrastive learning studies for negative samples [6].
>
> 2) As we employed a projection layer to map raw data into embeddings before applying data augmentation, such mapping disrupts the periodicity of the raw data.
>
> 3) Similar to sample-level contrasting, there is no assurance that all negative samples are exclusively hard-negatives (true negatives). It is quite common for certain negative samples to share the same label as the anchor sample [9]. In other words, there is no certainty, nor necessary, that all negative observations are dissimilar to the anchor observation.

---

> > ### Comment · Reviewer_Trt3 · 2023-08-17
> >
> > Thank the authors for the detailed response as well as the added experiments, which have answered part of my concerns. However, first, I do not agree with authors that "the majority of real-world applications of medical time series for disease diagnosis are binary classification tasks". In fact, real-world medical applications must be able to handle multi-class classification. Secondly, I would suggest the authors to refine their notations. In particular, it is quite confusing to use the same notation x to indicate both observation, sample, and trial.  Thirdly, both the sample-level consistency and the trail-level consistency are comparing at sample level. The difference lies in how the triplets are generated. Why the former use InfoNCE loss while the later use NT-Xent loss? The decision seems to be quite ad hoc. Given the above, I have raised my rating to 4.

---

> > > ### Author Response · Authors · 2023-08-18
> > > **Further explanations for your three comments**
> > >
> > > We thank the reviewer for the active feedback and give us the opportunity to further explain to your concerns.
> > >
> > > **New Q1**: I do not agree with authors that "the majority of real-world applications …  are binary classification tasks". In fact, real-world medical applications must be able to handle multi-class classification.
> > >
> > > **New A1**:
> > > 1.	We highly agree with the reviewer that real-world applications should be able to handle multi-class classification. Although the current dominant trend of AI research for medical time series centers around simple binary tasks [1-4], yet complexity of medical data and the diversity of clinical applications call for more comprehensive approaches.
> > >
> > > 2.	In response to your concern, we have demonstrated the efficacy of our model in addressing multi-classification challenges. We have conducted an additional experiment over a large-scale ECG dataset (PTB-XL, 17596 patients, 191400 samples) with 5 classes (See A4 and R-table 2).
> > >
> > > 3.	We acknowledge it's helpful to have more studies and analyses in terms of multi-class tasks. Will discuss it in our limitations of this paper, and list it as an important future work.
> > >
> > > [1] Liang, H., et al., 2019. Evaluation and accurate diagnoses of pediatric diseases using artificial intelligence. Nature medicine,.
> > >
> > > [2] Hicks, S.A., et al., 2022. On evaluation metrics for medical applications of artificial intelligence. Scientific reports.
> > >
> > > [3] Buch, V.H., et al., 2018. Artificial intelligence in medicine: current trends and future possibilities. British Journal of General Practice.
> > >
> > > [4] Soenksen, L.R., et al. 2022. Integrated multimodal artificial intelligence framework for healthcare applications. NPJ digital medicine.
> > >
> > > **New Q2**: Refine their notations. In particular, it is quite confusing to use the same notation x to indicate both observation, sample, and trial.
> > >
> > > **New A2**:
> > > We appreciate your attention to the notations used in our work.
> > >
> > > First, we clarify we didn’t use the same notation to indicate all of observation, sample, and trial. In this paper, our $x$ consistently represents a sample, while $r$ and $p$ stand for trial and patient, respectively.
> > >
> > > -	Within the sample level, $x_i$ denotes a sample with index $i$, and in the observation level, $x_{i, t}$ signifies an observation within sample $x_i$ at timestamp $t$. The subscripts $_i$ and $_t$ denote the $i$-th sample and the $t$-th observation.
> > >
> > > -	However, at the patient and trial levels, the subscript $i$ is unnecessary as the sample index doesn't matter here because we mainly care about how the positive/negative sample pairs are formulated. Here, the focus shifts to whether the sample $x$ originates from the same patient or trial. So, to simplify the notations, we omitted the subscript $_i$ in patient and trial levels, Instead, we use superscript $^+$ and $^-$ to denote the positive and negative samples, respectively, to emphasize the sample pair construction is more important than sample index.
> > >
> > > Second, we struggled in designing our paper's notation for a long time. We explored various plans, such as using different subscripts and superscripts to represent different levels (e.g., let $x_{s,o}^{p,t}$ denote the $o$-th observation from the $s$-th sample of the $t$-th trial of the $p$-th patient). However, we found such notations are overly complex and difficult to comprehend. Consequently, we opted for a simpler notation scheme (i.e., the one we presented in the paper) to make the whole model easier to follow.
> > >
> > > Your attention to our notation design is greatly valued, and we truly welcome any suggestions and advice to help us further polish the notations. We remain committed to refining our notations to ensure they are accessible and intuitive.
> > >
> > >
> > > **New Q3**: ....Why the former use InfoNCE loss while the later use NT-Xent loss? ...
> > >
> > > **New A3**: This is the Q8 of your original comments, which we have answered in rebuttal.  Due to the space limitation in response, we have presented the answers to your Q7-Q10 in the global response (prior to the references). For your convenience, we paste our answer here:
> > >
> > > - We observed that TS2Vec performs effectively for observation and sample levels. Concurrently, CLOCS introduced contrastive learning on ECG data, utilizing patient-level information, which bears similarity to our implementation at the trial level. Thus, we adopted their loss functions as a foundation for designing our own loss functions across distinct granularities.
> > >
> > > - Moreover, we clearly understand InfoNCE and NT-Xent are all derived from NCE loss family. We conducted preliminary experiments to compare using InfoNCE at four granularities, using NT-Xent in all granularities, and exchangeable use InfoNCE or NT-Xent at different granularities. The experimental results show InfoNCE and NT-Xent will lead to very similar outcomes while the NT-Xent can converge a bit faster. We will discuss this in our limitations and future work.

---

### Author Rebuttal · Authors · 2023-08-10

We appreciate the valuable feedback provided by all the reviewers. We highly appreciate the reviewers who believe our work is solid, effective, easy to follow, and well-written, with a logical presentation of the methodological contribution of the proposed method. In response to the reviewers' insightful comments, we have conducted additional experiments, including two new ablation studies, incorporating two new baselines, and utilizing a new dataset. We addressed your concerns and suggestions in the individual responses. We will incorporate the newly added content into the final version of the paper for camera-ready submission.

Thank you again for your thoughtful comments. We work hard to refine our paper, and we sincerely hope our responses are informative and helpful. If you feel we have not sufficiently addressed your concerns to motivate increasing your score, we would love to hear from you further on what points of concern remain and how we to improve our work. Thank you again!

Due to space limitations, we have included **some responses, all the tables, and all the references** within this global rebuttal. The experiment results of the ablation study in R-tables 1 and 2 can be found in the attached PDF file. Here is R-table 3.
***
**R-table 3: Comparison of the new large-scale dataset PTB-XL with new baseline CLOCS, NCL, and best old baseline TS2Vec.**
|   Label Fraction    |   Method    | Accuracy |  F1 |  AUROC |  AUPRC  |
| ----------- | ----------- | ----------- | ----------- | ----------- | ----------- |
| 100%  | CLOCS | 72.88±0.35 |  59.97±0.66|  88.30±0.17| 62.79±0.20|
| 100%  | NCL | 72.63±0.09 | 61.16±0.39 | 89.28±0.00 | 65.17±0.45 |
| 100%  | TS2Vec | 72.48±0.39 | 60.50±0.20 | 89.18±0.17 | 64.84±0.58 |
| 100%  | COMET(Ours) | **74.18±0.02** | **61.63±0.06** | **89.29±0.11** | **66.12±0.52** |
| 10%  | CLOCS | 67.00±0.82 |  54.81±0.36 |  83.66±0.45| 55.97±0.63|
| 10%  | NCL | 68.12±0.04 | 56.62±0.39 | 86.02±0.12 | 60.17±0.47 |
| 10%  | TS2Vec | 69.21±0.48 | 57.29±1.23 | **86.55±0.09** | 61.12±0.41 |
| 10%  | COMET(Ours) | **70.69±0.19** | **57.80±0.23** | 86.37±0.25 | **61.78±0.10** |
| 1%  | CLOCS | 52.16±2.35 |  36.39±2.43 |  69.68±1.85 | 35.48±2.13 |
| 1%  | NCL | 49.92±1.13 | 31.96±0.64 | 68.99±0.36 | 33.75±0.73 |
| 1%  | TS2Vec | 54.76±0.69 | 38.93±1.66 | 74.47±0.35 | 39.60±0.09 |
| 1%  | COMET(Ours) | **60.38±0.21** | **43.67±0.65** | **77.59±0.75** | **45.75±0.93** |

***
## Here are some continued responses to reviewer **Trt3**'s questions.
***
**Q7** Would jittering and masking break the continuity of the normal time series data?

**A7** We respectively clarify that we did not use jittering ( Line 196 just serves as an example). Our experiments solely used masking(Sec. 5 implementations; Appendix B data augmentation bank). As previously mentioned in A2, we carefully designed the strength of data augmentation (10% masking ratio) to maximize the preservation of features within the time series, which will not distort the data continuity.
***

**Q8** Explain the usage of InfoNCE at observation and sample blocks while using NT-Xent loss at trial and patient levels.

**A8** We observed that TS2Vec performs effectively for observation and sample levels. Concurrently, CLOCS introduced contrastive learning on ECG data, utilizing patient-level information, which bears similarity to our implementation at the trial level. Thus, we adopted their loss functions as a foundation for designing our own loss functions across distinct granularities. Broadly speaking, these loss functions are all variants of the InfoNCE loss function.
***
**Q9** SOTA comparison: why NCL and CLOCS are not considered?

**A9** We did test running for CLOCS before submission and found it extremely time-consuming (More than 10 hrs for the PTB dataset in a single run). As a result, we chose some other SOTAs as baselines. Regarding NCL, their definition of neighborhood contrasting inspired us to design trial level contrasting. We compare with CLOCS and NCL on the newly added large-scale dataset PTB-XL. See R-table 3.
***
**Q10** Any ablation study on the hyperparameters $\lambda_i$ is missing.

**A10** We add ablation study on $\lambda_i$. See R-table 1 and 2.
***
# Reference

[1] Kiyasseh, Dani., et al. "Clocs Contrastive learning of cardiac signals across space, time, and patients." ICML, 2021.

[2] Lan, Xiang., et al. "Intra-inter subject self-supervised learning for multivariate cardiac signals."  AAAI, 2022.

[3] Banville, Hubert., et al. "Uncovering the structure of clinical EEG signals with self-supervised learning." Journal of Neural Engineering, 2021

[4] Tonekaboni., et al. "Unsupervised representation learning for time series with temporal neighborhood coding." arXiv 2021

[5] Ieracitano., et al. "A convolutional neural network approach for classification of dementia stages based on 2D-spectral representation of EEG recordings." Neurocomputing, 2019

[6] Yue, Zhihan., et al. "Ts2vec Towards universal representation of time series." AAAI, 2022.

[7] Tzimourta, Katerina D., et al. "Machine learning algorithms and statistical approaches for Alzheimer’s disease analysis based on resting-state EEG recordings A systematic review." Inter. J. of Neural Syst, 2021

[8] Zhang, Xiang, and Lina Yao. Deep Learning for EEG-Based Brain–Computer Interfaces Representations, Algorithms and Applications. 2021.

[9] Kalantidis, Yannis., et al. "Hard negative mixing for contrastive learning." Neurips, 2020

[10] Xinyu Yang., et al. “A self-supervised contrastive learning framework for univariate time series representation.” Knowledge-Based Systems, 2022.

[11] Machine Learning Mastery. "Introduction to Regularization to Reduce Overfitting and Improve Generalization Error."  https://machinelearningmastery.com/introduction-to-regularization-to-reduce-overfitting-and-improve-generalization-error/

[12] TensorFlow. "Overfit and Underfit." https://www.tensorflow.org/tutorials/keras/overfit_and_underfit

---

### Comment · Area_Chair_SinJ · 2023-08-15
**please engage in discussion**

Hello reviewers.

It looks like right now the reviews are a bit divergent. If you have not already done so, please read the authors' response to your reviews *and to the other reviewers' reviews*, and discuss. The authors have provided reasonably detailed rebuttals so it would be great if you engage in discussion as soon as possible, especially to indicate if your opinion of the paper has changed or if you would like any sort of additional comments or clarifications.

Thanks,
Area chair

---

### Decision · Program_Chairs · 2023-09-21

**Decision:**

Accept (poster)

**Comment:**

Overall, the authors provided very detailed responses to address reviewer concerns in a satisfactory manner, which also involved revising the paper to significantly improve it during the author-reviewer discussion period (with extensive clarifications and also more experimental results including more baselines). Three out of the four reviewers at this point lean on acceptance of the paper including one favoring strong acceptance. The only dissenting reviewer voted for a borderline reject but it seems like the authors have sufficiently addressed this reviewer's concerns.